# ND-SDF: Learning Normal Deflection Fields for High-Fidelity Indoor Reconstruction

**Ziyu Tang[1], Weicai Ye[1,2,✉], Yifan Wang[2], Di Huang[2], Hujun Bao[1], Tong He[2,✉], Guofeng Zhang[1]**
[1]State Key Lab of CAD&CG, Zhejiang University, [2]Shanghai AI Laboratory
https://zju3dv.github.io/nd-sdf/

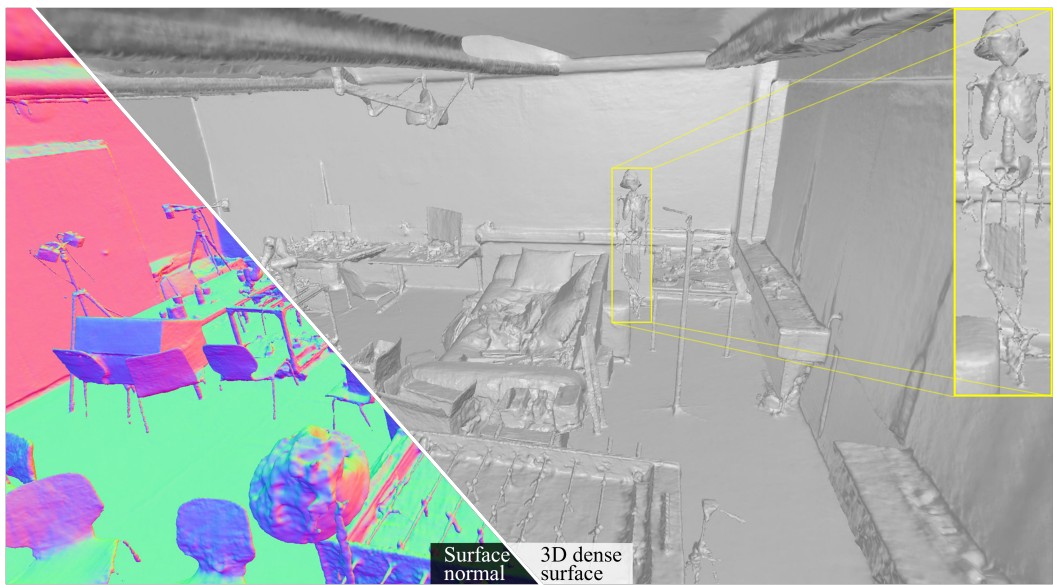

Figure 1: We present **ND-SDF**, a framework for high-fidelity 3D indoor surface reconstruction from multi-views. Shown above is an extracted mesh from ScanNet++.

## ABSTRACT

Neural implicit reconstruction via volume rendering has demonstrated its effectiveness in recovering dense 3D surfaces. However, it is non-trivial to simultaneously recover meticulous geometry and preserve smoothness across regions with differing characteristics. To address this issue, previous methods typically employ geometric priors, which are often constrained by the performance of the prior models. In this paper, we propose **ND-SDF**, which learns a Normal Deflection field to represent the angular deviation between the scene normal and the prior normal. Unlike previous methods that uniformly apply geometric priors on all samples, introducing significant bias in accuracy, our proposed normal deflection field dynamically learns and adapts the utilization of samples based on their specific characteristics, thereby improving both the accuracy and effectiveness of the model. Our method not only obtains smooth weakly textured regions such as walls and floors but also preserves the geometric details of complex structures. In addition, we introduce a novel ray sampling strategy based on the deflection angle to facilitate the unbiased rendering process, which significantly improves the quality and accuracy of intricate surfaces, especially on thin structures. Consistent improvements on various challenging datasets demonstrate the superiority of our method.

# 1 INTRODUCTION

3D surface reconstruction Chen et al. (2024a); Wang et al. (2024); Ye et al. (2022; 2023b); Liu et al. (2021); Li et al. (2020) aims to recover watertight, dense 3D geometry from multi-view images Hartley & Zisserman (2003), representing a significant research area in computer vision and graphics. The recovered surfaces have proven invaluable for many downstream tasks, including robotic navigation, AR/VR, and smart cities.

Recently, coordinate-based networks Mildenhall et al. (2021); Barron et al. (2022); Müller et al. (2022); Martin-Brualla et al. (2021); Zhang et al. (2020); Ye et al. (2023a); Huang et al. (2024); Ming et al. (2022) are drawing increasing attention, due to their remarkable performance in the task of novel view synthesis. Inspired by implicit representations of the scene, many subsequent works have introduced Signed Distance Functions (SDF) Park et al. (2019) or occupancy Li et al. (2022) to parameterize the 3D geometry. Notably, some of these approaches Yariv et al. (2021); Wang et al. (2021) outperform other methods in recovering dense surfaces by representing scene geometry with implicit distance fields and integrating them with differentiable rendering frameworks to enhance surface reconstruction.

However, recovering high-fidelity surfaces remains challenging, as relying solely on color images for supervision often results in an underconstrained problem, especially for regions like textureless walls and ceilings. Previous methods Yu et al. (2022b); Wang et al. (2022) have attempted to mitigate this issue by incorporating auxiliary supervision. For instance, MonoSDF employs monocular cues from a pretrained model as pseudo-ground truth to supervise the reconstruction process, which alleviates the issue in textureless regions. However, the large errors due to domain gaps in monocular cues and the inconsistencies in view-dependent prior guidance often lead to visible detail loss and erroneous surfaces Yu et al. (2022b).

Noting that prior models are highly accurate in simpler regions like floors and walls but struggle with complex geometries. Based on this insight, we introduce ND-SDF for high-fidelity indoor reconstruction (Figure 1). Our core innovation is the construction of a deflection field that adaptively learns the geometric deviations between the actual scene geometry and the prior geometry derived from normal priors. The inherent deviation is defined as the angular difference between the true scene normals and the prior normals. By aligning the deflected scene normals with the normal cues, our model can adaptively learn the deviation, which encourages accurate recovery of intricate structures maintaining fine details without being misled by erroneous priors.

In addition, we propose a novel adaptive deflection angle prior loss that leverages prior normals for differential supervision of high and low-frequency areas, achieving an optimal balance between smoothness and detail. Our observations indicate that large angle deviations are primarily concentrated in thin and fine-grained structures. Building on this insight, we introduce deflection-angle guided optimization to proactively facilitate the recovery of detailed structures. Furthermore, we address a significant bias issue and propose deflection angle guided unbiased rendering to improve the reconstruction of small or thin structures, as shown in Figure 1. In summary, we present the following contributions:

- We propose a novel scene attribute field, named normal deflection field, which adaptively learns the deviations between the scene and normal priors. This method aids us in distinguishing between detailed and textureless areas. Therefore, we can restore more fine structures while ensuring the smoothness and integrity of the scene.

- Empirically assuming that prior models incur larger errors in complex areas, we employ the deflection angle to discriminate between high and low-frequency regions. Consequently, we propose a novel adaptive deflection angle prior loss that dynamically adjusts the utilization of distinct cues, thereby achieving a balance between complex structures and smooth surfaces. Furthermore, we utilize the deflection angle to guide ray sampling and photometric optimization, facilitating the restoration of finer-grained structures.

- To address the inherent bias issue of neural surface rendering, we integrate the unbiasing method Zhang et al. (2023) to implement an adaptive, unbiased rendering strategy. This approach facilitates the recovery of extremely thin structures without compromising scene fidelity. Our method outperforms previous approaches significantly in indoor reconstruction evaluations, and this superiority is validated through extensive ablation experiments.

## 2 RELATED WORK

**Neural Surface Representation of 3D scenes**   Recently, the representation of 3D scenes using neural fields has gained popularity due to their expressiveness and simplicity. NeRF-type approaches Mildenhall et al. (2021); Barron et al. (2022); Müller et al. (2022); Fridovich-Keil et al. (2022); Sun et al. (2022) have proposed encoding coordinate-based density and appearance of scenes by utilizing simple multilayer perceptrons (MLPs) and explicit structures such as voxel grids, resulting in photorealistic novel view synthesis. However, these approaches fail to accurately recover surfaces due to the lack of constraints on density. To address this issue, subsequent works, such as VolSDF Yariv et al. (2021) and Neus Wang et al. (2021), implicitly represent signed distance functions (SDFs) and employ a SDF-density conversion function to encourage precise surface reconstruction. Other methods Li et al. (2023b); Rosu & Behnke (2023); Wang et al. (2023); Yariv et al. (2023); Fu et al. (2022); Zhang et al. (2023) have also been proposed, introducing different representations and optimization techniques to further enhance reconstruction quality and efficiency. Nevertheless, these aforementioned methods struggle to handle indoor scenes with a large number of low-frequency areas (e.g., walls and floors), as photometric loss becomes unreliable in such regions.

**Neural reconstruction in Indoor Environments**   Due to the complex layouts of indoor scenes, additional auxiliary data are required for reasonable reconstruction. Manhattan-SDF Guo et al. (2022) employs the Manhattan assumption and semantic priors to jointly regularize textureless regions such as walls and floors. HelixSurf Liang et al. (2023) achieves intertwined regularization iteratively by combining neural implicit surface learning with PatchMatch-based multi-view stereo (MVS) Barnes et al. (2009) robustly and efficiently. Other works Yu et al. (2022b); Wang et al. (2022) propose utilizing monocular priors from pretrained models to achieve smooth and complete reconstruction. However, directly applying normal or depth priors Yu et al. (2022b) may lead to undesirable reconstruction results due to the unreliable nature of these priors. NeuRIS Wang et al. (2022) filters out unreliable priors based on the assumption that areas with rich 2D visual features are more error-prone. H2O-SDF Park et al. (2024) simply utilizes uncertainty prior to re-weight the normal prior loss. Nevertheless, both of these methods exhibit weak generalization capabilities. For example, the visual feature hypothesis Wang et al. (2022) struggles to differentiate between rich-textured planar areas. Similarly, prior-guided re-weighting Park et al. (2024) is further constrained by the domain gap between the prior model and the scene. On the other hand, DebSDF Xiao et al. (2024) effectively designs an uncertainty field based on a probabilistic model to guide the loss function, leading to more robust and accurate reconstructions. In contrast to methods with weaker generalization capabilities and those relying on probabilistic models, our proposed method, ND-SDF, learns a Normal Deflection field that enables the dynamic adaptation of priors based on their characteristics. This field directly models geometrically meaningful SO(3) residuals, resulting in highly detailed surface reconstructions while maintaining smoothness and robustness.

## 3 METHOD

The primary objective of our Normal Deflection Signed Distance Function (ND-SDF) is to reconstruct dense surfaces from calibrated multi-view images, with a primary focus on indoor scenes with established priors. To achieve this, we introduce a novel component known as the normal deflection field. This field is designed to quantify and learn the deviations between actual scene geometries and their corresponding normal priors. By integrating this field, our method can adaptively supervise both high and low-frequency regions in the scene. This adaptive supervision is crucial for preserving fine details while ensuring overall surface smoothness.

The operational framework of our approach is depicted in Figure 2. In subsequent sections, we delve into a comprehensive discussion on the implementation of the normal deflection field. Additionally, we explore various strategies that leverage this field to enhance the fidelity of surface details, thereby promoting a more accurate surface reconstruction.

### 3.1 PRELIMINARIES

**Volume Rendering** NeRF assumes that a ray $\mathbf{r}(t) = \mathbf{o} + t\mathbf{v}$ is emitted from viewpoint $\mathbf{o}$ in direction $\mathbf{v}$, where $t$ denotes the distance from the viewpoint. $N$ points are sampled along the ray, i.e. $\mathbf{x}_i =$

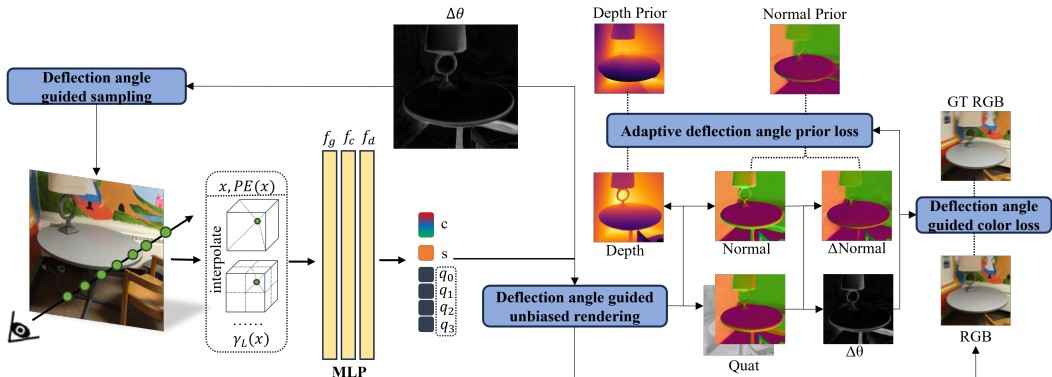

Figure 2: **Overview of Our method.** We utilize multi-resolution hash grids $\gamma_L$ as scene representation. The core of ND-SDF is the normal deflection field, which we represent using quaternions predicted by the deflection network (denoted as $f_d$). We align the deflected normals with the prior normals to learn the deviation between the scene and the priors. To distinctly supervise high and low-frequency areas, we employ an adaptive deflection angle prior loss, ensuring both smoothness and detail. Furthermore, we utilize the deflection angle $\Delta\theta$ to distinguish complex structures, enabling angle guided sampling and color loss to facilitate the recovery of intricate surface details. Lastly, we combine the unbiased rendering method Zhang et al. (2023) to ensure the generation of extremely thin structures indoors.

$\mathbf{o} + t_i\mathbf{v}, i \in \{1, 2...N\}$. Given each point's volume density $\sigma_i$ and color $\mathbf{c}_i$, the color of the ray $\hat{\mathbf{C}}(\mathbf{r})$ can be synthesized using volume rendering techniques as:

$$\hat{\mathbf{C}}(\mathbf{r}) = \sum_{i=1}^{N} T_i\alpha_i\mathbf{c}_i, \; \alpha_i = 1 - \exp\left(-\sigma_i\delta_i\right), T_i = \prod_{j=1}^{i-1}(1-\alpha_j), \tag{1}$$

where $\alpha_i$ is the opacity of $i$-th ray segment, $T_i$ denotes the transmittance of light reaching the $i$-th ray segment, $\delta_i = t_i - t_{i-1}$ represents the distance between adjacent sample points.

A color loss supervised using ground-truth (GT) color is utilized to optimize the neural fields:

$$\mathcal{L}_{\text{color}} = \sum_{\mathbf{r} \in \mathcal{R}} \|\hat{\mathbf{C}}(\mathbf{r}) - \mathbf{C}(\mathbf{r})\|_1, \tag{2}$$

where $\mathcal{R}$ is the sampled ray set, $\mathbf{C(r)}$ is the GT color.

**SDF-induced Volume Rendering** The rise of neural surface reconstruction is attributed to seminal works like VolSDF and NeuS, which propose specific SDF to volume density conversion, allowing optimization via volume rendering. The final surface, denoted as $S$, is equivalent to the zero level-set of the implicit distance field, i.e., $S = \{\mathbf{x} \in \mathbb{R}^3 | s(\mathbf{x}) = 0\}$, where $s(\mathbf{x})$ is the SDF value.

We follow VolSDF Yariv et al. (2021), employing the Laplace Cumulative Distribution Function (CDF) to model the relationship between SDF and volume density:

$$\sigma(\mathbf{x}) = \frac{1}{\beta}\Psi_\beta\left(-s(\mathbf{x})\right) = \begin{cases} \frac{1}{2\beta}\exp\left(\frac{-s(\mathbf{x})}{\beta}\right) & \text{if } s(\mathbf{x}) \geq 0 \\ \frac{1}{\beta} - \frac{1}{2\beta}\exp\left(\frac{s(\mathbf{x})}{\beta}\right) & \text{if } s(\mathbf{x}) < 0 \end{cases}, \tag{3}$$

where $\Psi_\beta$ denotes the Laplace CDF, $\beta$ denotes the variance. Depth $\hat{D}(\mathbf{r})$ and normal $\hat{\mathbf{N}}(\mathbf{r})$ at the surface intersection are synthesized using the same approach as that employed for synthesizing colors:

$$\hat{D}(\mathbf{r}) = \sum_{i=1}^{N} T_i\alpha_i t_i, \hat{\mathbf{N}}(\mathbf{r}) = \sum_{i=1}^{N} T_i\alpha_i\mathbf{n}_i. \tag{4}$$

Here, $\mathbf{n}_i$ is the analytical gradient of the SDF network at point $i$. Following MonoSDF Yu et al. (2022b), we utilize monocular depth and normal priors generated from pretrained model Eftekhar et al. (2021) to supervise the rendered depth and normal:

$$\begin{aligned} \mathcal{L}_{\text{depth}} &= \sum_{\mathbf{r} \in \mathcal{R}} \|w\hat{D}(\mathbf{r}) + q - D(\mathbf{r})\|^2 \\ \mathcal{L}_{\text{normal}} &= \sum_{\mathbf{r} \in \mathcal{R}} \|\hat{\mathbf{N}}(\mathbf{r}) - \mathbf{N}(\mathbf{r})\|_1 + \left\|1 - \hat{\mathbf{N}}(\mathbf{r})^T\mathbf{N}(\mathbf{r})\right\|_1 \end{aligned}, \tag{5}$$

where the two coefficients $(w, q)$ obtained by least square algorithms are utilized to align the scale between monocular depth and rendered depth, i.e., $w\hat{D} + q \approx D$.

We encode scene geometry using Instant-NGP Müller et al. (2022) $\gamma_L$ and a shallow MLP $f_g$, that is, $(s(\mathbf{x}) \in \mathbb{R}^3, \mathbf{z}(\mathbf{x}) \in \mathbb{R}^{256}) = f_g(\mathbf{x}, PE(\mathbf{x}), \gamma_L(\mathbf{x}))$, where $\mathbf{z}(\mathbf{x})$ denotes the latent geometry feature.

Also, we utilize the eikonal term Gropp et al. (2020); Yariv et al. (2020) to regularize the shape of SDF in 3D space:

$$\mathcal{L}_{\text{eik}} = \frac{1}{N} \sum_{i=1}^{N} \left( \|\nabla s(\mathbf{x}_i)\|_2 - 1 \right)^2. \tag{6}$$

Following Neuralangelo Li et al. (2023b), we further employ numerical gradients to enhance surface geometric consistency and use curvature loss for smoothness. The detailed definitions of them are provided in the appendix.

### 3.2 NORMAL DEFLECTION FIELD

A significant challenge in indoor 3D reconstruction is achieving a balance between the smoothness of overall surfaces and the intricacy of complex structures. Traditional approaches relying solely on photometric loss have proven inadequate for accurately capturing smooth areas, particularly in texture-deficient regions. Later methods often leverage auxiliary data, such as normal priors, to enhance the reconstruction quality in textureless areas. However, the uniform application of normal priors across different scene types can impair the recovery of complex structures. This is primarily due to the variable reliability of these priors in diverse regions, where they may not accurately reflect the underlying geometry. To address this issue, we propose the development of a Normal Deflection Field. This field is designed to dynamically represent the existing 3D angular deviation between the actual scene normals and the provided normal priors. By doing so, it effectively circumvents potential misguidance arising from inconsistent priors, thereby enabling a more reliable and nuanced reconstruction of both smooth and complex indoor structures.

Specifically, we choose quaternion as the deflection form, which is a lightweight rotation representation. The quaternion can be parameterized by a single MLP $f_d$:

$$\mathbf{q}_i = f_d(\mathbf{x}_i, \mathbf{v}_i, \mathbf{n}_i, \mathbf{z}_i), \tag{7}$$

where $\mathbf{q}_i = (\mathbf{q}_i^0, \mathbf{q}_i^1, \mathbf{q}_i^2, \mathbf{q}_i^3)$ is the deflection quaternion (normalized default) for the sampled $\mathbf{x}_i$. The quaternion at the surface intersection point is synthesized using volume rendering techniques, analogous to the way NeRF synthesizes colors:

$$\mathbf{Q}(\mathbf{r}) = \sum_{i=1}^{N} T_i \alpha_i \mathbf{q}_i. \tag{8}$$

We deflect the rendered normal using the synthesized deflection quaternion $\mathbf{Q}(\mathbf{r})$:

$$\hat{\mathbf{N}}^d(\mathbf{r}) = \mathbf{Q}(\mathbf{r}) \otimes \hat{\mathbf{N}}(\mathbf{r}) \otimes \mathbf{Q}^{-1}(\mathbf{r}), \tag{9}$$

where $\hat{\mathbf{N}}^d$ denotes the deflected rendered normal, $\mathbf{Q}^{-1}$ denotes the inverse of $\mathbf{Q}$, also known as the conjugate, and $\otimes$ is a quaternion multiplication operation. Since a quaternion can be represented in trigonometric form, i.e. $\mathbf{Q} = \cos\frac{\theta}{2} + \sin\frac{\theta}{2}(\mathbf{u}^1 i, \mathbf{u}^2 j, \mathbf{u}^3 k)$, this operation is to rotate the rendered normal around the quaternion axis $\mathbf{u} = (\mathbf{u}^1, \mathbf{u}^2, \mathbf{u}^3)$ by $\theta$ degrees.

The deviation between the scene and priors is learned by minimizing the difference between the deflected rendered normal and the prior normal. Thus, we define the deflected normal loss:

$$\mathcal{L}_{normal}^d = \sum_{\mathbf{r} \in \mathcal{R}} \|\hat{\mathbf{N}}^d(\mathbf{r}) - \mathbf{N}(\mathbf{r})\|_1 + \left\| 1 - \hat{\mathbf{N}}^d(\mathbf{r})^T \mathbf{N}(\mathbf{r}) \right\|_1. \tag{10}$$

### 3.3 ADAPTIVE DEFLECTION ANGLE PRIOR LOSS

We assume that the learned deviation is encompassed within the actual deflected angle when the scene normal is deflected. This deflected angle $(\Delta\theta)$ is computed as follows:

$$\Delta\theta = \arccos\left(\hat{\mathbf{N}}(\mathbf{r}) \cdot \hat{\mathbf{N}}^d(\mathbf{r})\right), \tag{11}$$

where $\Delta\theta \in [0, \pi]$. Empirically, the deviation between the scene and priors increases as the structure becomes more complex. This intuitively suggests that the deflection angle is small in smooth regions and large in high-frequency zones. Based on this, the adaptive deflection angle normal prior loss is proposed to dynamically adjust the utilization of differing priors based on their characteristics:

$$\mathcal{L}_{normal}^{ad} = \sum_{\mathbf{r}\in\mathcal{R}} g^d(\Delta\theta)\mathcal{L}_{normal}^d(\hat{\mathbf{N}}^d(\mathbf{r}), \mathbf{N}(\mathbf{r})) + g(\Delta\theta)\mathcal{L}_{normal}(\hat{\mathbf{N}}(\mathbf{r}), \mathbf{N}(\mathbf{r})) \ , \qquad (12)$$

where $g^d$ and $g$ are modulation functions that adjust the weights of the deflected and original normal loss terms based on the deflection angle. We similarly define an adaptive depth prior loss:

$$\mathcal{L}_{depth}^{ad} = \sum_{\mathbf{r}\in\mathcal{R}} g(\Delta\theta)\mathcal{L}_{depth}(\hat{D}(\mathbf{r}), D(\mathbf{r})). \qquad (13)$$

As $\Delta\theta$ increases, $g^d(\Delta\theta)$ increases, and $g(\Delta\theta)$ decreases, since we should less apply priors in complex areas indicated by large deflection angles. The combined adaptive normal and depth prior loss is termed the adaptive deflection angle prior loss (as illustrated in Figure 2); see appendix for detailed definitions of the proposed modulation functions.

## 3.4 Deflection angle guided optimization

Through the utilization of the proposed adaptive prior loss, we dynamically adjust the utilization of various priors, significantly enhancing the quality of reconstruction. However, this alone is insufficient for generating more complex structures. Fundamentally, merely learning the deviation does not endow the capability to reconstruct additional details. Recognizing that large deflection angles indicate complex areas, we introduce three deflection angle guided optimization methods specifically designed to facilitate the recovery of more thin and fine-grained structures.

**Deflection angle guided sampling.** Textureless areas, such as walls, constitute a significant portion of indoor scenes. These regions converge rapidly by leveraging prior knowledge, partly due to their inherent simplicity. To prevent continuous oversampling of well-learned smooth regions, we proactively sample more rays in complex areas to capture finer details. The sampling process is naturally guided by the learned deflection angles (see Appendix A.3.2 for details).

**Deflection angle guided photometric loss.** To further enhance detail recovery, we impose additional photometric loss on complex areas identified by the deflection angles. The original color loss term is re-weighted based on the deflection angle, resulting in a re-weighted color loss.

$$\mathcal{L}_{color}^d = \sum_{\mathbf{r}\in\mathcal{R}} w_{color}(\Delta\theta(\mathbf{r}))\|\hat{\mathbf{C}}(\mathbf{r}) - \mathbf{C}(\mathbf{r})\|_1. \qquad (14)$$

$\Delta\theta(\mathbf{r})$ denotes the actual deflected angle and $w_{color}$ is the re-weighting function (Appendix A.3.2).

**Deflection angle guided unbiased rendering.** Accurately reconstructing thin structures, such as chair legs, remains challenging. This limitation arises from the inherent bias issues in SDF-induced volume rendering, manifesting in two ways: (1) According to TUVR Zhang et al. (2023), the derivative of the rendering weight $\frac{\partial W(t)}{\partial t}$ is influenced by various angle differences between ray directions and scene normals. This results in a non-maximum value of $w$ at the surface intersection point, thereby degrading the surface quality; (2) For rays passing close to the object, the Laplace CDF assigns a high density to points near the surface, which leads to incorrect rendering weights, and subsequently affects the depth and normal compositing. This issue causes thin structures to disappear during training. Therefore, it is crucial to apply an unbiased rendering method during reconstruction. We illustrate this biased scenario in Fig. 3, highlighting the significance of ensuring unbiasedness.

Inspired by TUVR Zhang et al. (2023), we transform SDF to density using an unbiased function:

$$\sigma(\mathbf{r}(t_i)) = \frac{1}{\beta}\Psi_\beta\left(\frac{-f_g(\mathbf{r}(t_i))}{|f_g'(\mathbf{r}(t_i))|}\right), \qquad (15)$$

where $\mathbf{r}(t_i) = \mathbf{o} + t_i\mathbf{v}$ is a sampled point on the ray, and $f_g$ is the geometry network, which predicts the SDF value. This transformation effectively mitigates bias, as shown in Fig 3. However, we observe that simply applying it leads to poor convergence of the overall surface. Therefore, we only partially apply it to thin structures indicated by the learned deflection angles. A detailed analysis of the convergence issue is presented in Appendix B.1.

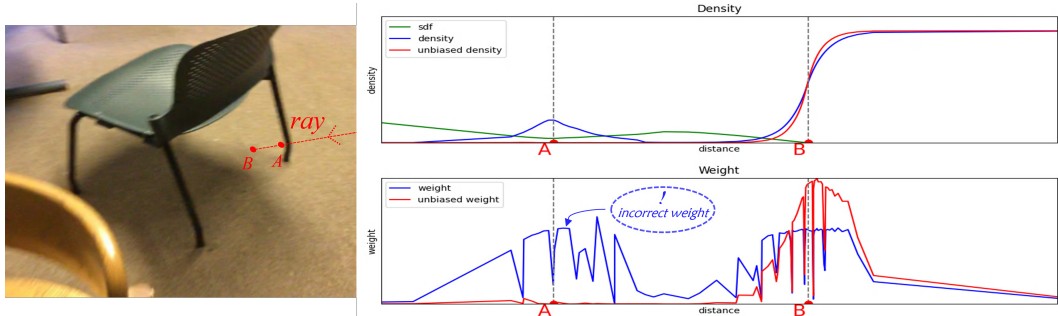

Figure 3: We present a scenario in which a ray passes close to a chair leg. Along the ray, point A is situated near the chair leg, while point B is located at the ray-ground intersection. In the absence of unbiasing, abnormal density and rendering weight are observed at point A, whereas density should be exclusively assigned to point B.

| Method | Acc↓ | Comp↓ | Pre↑ | Recall↑ | Chamfer↓ | F-score↑ |
|---|---|---|---|---|---|---|
| COLMAP Schönberger et al. (2016) | 0.047 | 0.235 | 71.1 | 44.1 | 0.141 | 53.7 |
| VolSDF Yariv et al. (2021) | 0.414 | 0.120 | 32.1 | 39.4 | 0.267 | 34.6 |
| Neus Wang et al. (2021) | 0.179 | 0.208 | 31.3 | 27.5 | 0.194 | 29.1 |
| NeuRIS Wang et al. (2022) | 0.050 | 0.049 | 71.7 | 66.9 | 0.050 | 69.2 |
| MonoSDF Yu et al. (2022b) | 0.035 | 0.048 | 79.9 | 68.1 | 0.042 | 73.3 |
| HelixSurf Liang et al. (2023) | 0.038 | 0.044 | 78.6 | 72.7 | 0.042 | 75.5 |
| DebSDF Xiao et al. (2024) | 0.036 | 0.040 | 80.7 | 76.5 | 0.038 | 78.5 |
| H2OSDF Park et al. (2024) | 0.032 | 0.037 | 83.4 | 76.9 | 0.035 | 79.9 |
| Ours | **0.031** | **0.036** | **84.0** | **80.3** | **0.034** | **82.0** |

Table 1: **Quantitative results on the ScanNet dataset.** Our method substantially outperforms previously best-performing methods across all metrics to date.

## 3.5 OPTIMIZATION

The overall loss function is defined as:

$$\mathcal{L} = \mathcal{L}_{color}^{d} + \lambda_1 \mathcal{L}_{eik} + \lambda_2 \mathcal{L}_{curv} + \lambda_3 \mathcal{L}_{normal}^{ad} + \lambda_4 \mathcal{L}_{depth}^{ad}. \tag{16}$$

The weights $\lambda_1...\lambda_4$ are utilized to balance the importance of these loss terms.

## 4 EXPERIMENT

**Datasets** We conducted experiments on four indoor datasets: ScanNet Dai et al. (2017), Replica Straub et al. (2019), TanksandTemples Knapitsch et al. (2017), and ScanNet++ Yeshwanth et al. (2023). ScanNet contains 1513 indoor scenes captured with an iPad, processed using the BundleFusion algorithm to obtain camera poses and surface reconstruction. Replica is a synthetic dataset comprising 18 indoor scenes. Each scene features dense geometry and high dynamic range textures. TanksandTemples is a large-scale 3D reconstruction dataset including high-resolution outdoor and indoor environments. We followed the splits from MonoSDF and applied the same evaluation settings. We also conducted experiments on ScanNet++, which includes 460 indoor scenes captured using laser scanners. These scenes offer high-quality dense reconstructions and images. For testing, we selected six scenes from ScanNet++.

**Implementation details** Our method was implemented using PyTorch Paszke et al. (2019). The image resolution for all scenes is 384×384. We obtained normal and depth cues using Omnidata Eftekhar et al. (2021). Multi-resolution hash grids were utilized for scene representation. Both the geometry network and color network consists of two layers, each with 256 nodes. Our network was optimized using AdamW Loshchilov & Hutter (2017) optimizer with a learning rate of 1e-3. The weights for loss terms were: $\lambda_1 = 0.05$, $\lambda_2 = 0.0005$, $\lambda_3 = 0.025$, $\lambda_4 = 0.05$. Upon adequate initialization of the deflection field, we initiated deflection angle guided sampling, photometric optimization, and unbiased rendering. All experiments were conducted on an NVIDIA TESLA A100 PCIe 40GB, with each iteration sampling 4×1024 rays, totaling 128,000 training steps. The training time was about 14 GPU hours. Our hash encoding resolution spanned from $2^5$ to $2^{11}$ across 16 levels, with the initial activation level set to 8 and activation steps set to 2000.

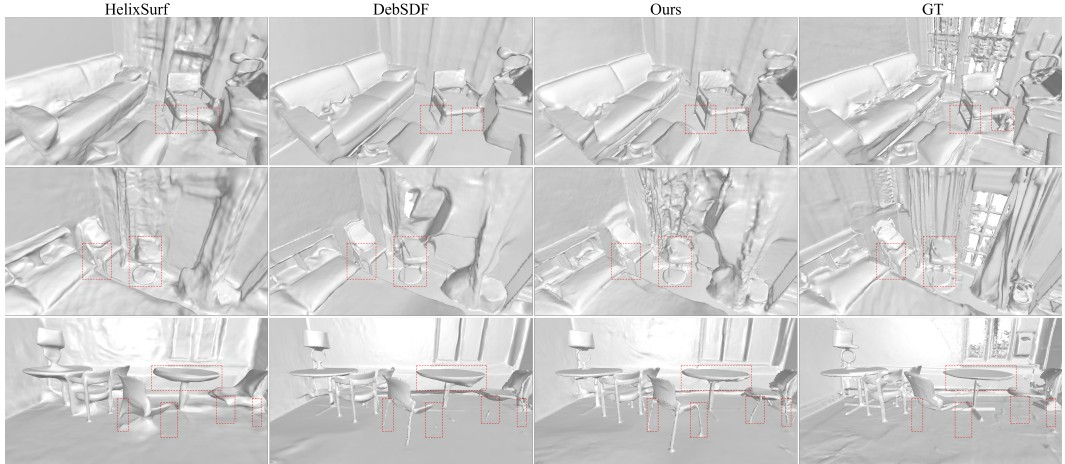

Figure 4: **Qualitative results on ScanNet.** Our approach effectively generates thin structures, such as chair legs, and produces more accurate surfaces.

| Method | Auditorium | Ballroom | Courtroom | Museum | Mean |
|---|---|---|---|---|---|
| NeuRIS Wang et al. (2022) | 0.79 | 0.84 | 2.78 | 1.34 | 1.13 |
| MonoSDF (Grid) Yu et al. (2022b) | 6.32 | 6.43 | 18.73 | 8.96 | 10.11 |
| DebSDF (Grid) Xiao et al. (2024) | **7.45** | 4.21 | 19.33 | 7.64 | 9.66 |
| Ours | 7.07 | **10.74** | **23.92** | **16.11** | **14.46** |

Table 2: **F-score on the TanksandTemples dataset**. 'Grid' indicates using hash encoding as the feature representation backend. More detailed comparison results can be found in the appendix.

**Metrics** In accordance with prior researches, we employ six standard metrics to evaluate the reconstructed meshes: Accuracy, Completeness, Chamfer Distance, Precision, Recall, and F-score. Additionally, normal consistency is used to evaluate the Replica dataset.

**Baselines** We compare our approach with the following methods: (1) traditional MVS techniques COLMAP Schonberger & Frahm (2016); Schönberger et al. (2016); (2) basic neural implicit methods, including VolSDF and NeuS; (3) methods incorporating auxiliary data, including MonoSDF and NeuRIS; (4) other indoor reconstruction methods, such as HelixSurf Liang et al. (2023).

## 4.1 COMPARISONS

**ScanNet:** In Figure 4, we present qualitative results and in Table 1, quantitative results are displayed. These quantitative results are averaged over the selected four scenes, consistent with MonoSDF. ND-SDF significantly surpasses previous best-performing works, achieving state-of-the-art performance. Specifically, we achieve the highest F-score, a reliable metric that reflects reconstruction accuracy by balancing both Accuracy and Completeness. When compared with HelixSurf and DebSDF (as shown in Figure 4), our approach also accurately captures thin structures, such as chair legs. The results highlight the effectiveness of our proposed deflection methods, which substantially enhance the surface accuracy and the recovery of thin and fine-grained structures in indoor environments.

| Method | Acc↓ | Comp↓ | Pre↑ | Recall↑ | Chamfer↓ | F-score↑ |
|---|---|---|---|---|---|---|
| VolSDF Yariv et al. (2021) | 0.070 | 0.102 | 0.405 | 0.339 | 0.086 | 0.368 |
| Baked-Angelo Yu et al. (2022a) | 0.152 | 0.039 | 0.543 | 0.718 | 0.095 | 0.614 |
| MonoSDF (Grid) Yu et al. (2022b) | 0.057 | 0.032 | 0.651 | 0.703 | 0.044 | 0.675 |
| DebSDF (Grid) Xiao et al. (2024) | 0.060 | 0.031 | **0.669** | 0.727 | 0.045 | 0.696 |
| Ours | **0.056** | **0.024** | 0.667 | **0.785** | **0.040** | **0.721** |

Table 3: **Quantitative results on the ScanNet++ dataset.** We select 6 scenes and calculate their average metrics. Baked-Angelo is the best-performing reconstruction method tested without priors.

**Replica:** Following MonoSDF Yu et al. (2022b), we used their pre-processed masks to filter out abnormal priors. Quantitative results are presented in Table 4, where our approach achieves the highest F-score and the lowest Chamfer Distance, significantly surpassing previous methods.

**Tanks and temples:** We also conducted experiments on the advanced split of T&T, featuring several challenging large-scale indoor scenes. As shown in Table 2, ND-SDF achieves the highest average F-

| Method | Normal C.↑ | Chamfer↓ | F-score↑ |
|---|---|---|---|
| Unisurf Oechsle et al. (2021) | 90.96 | 4.93 | 78.99 |
| MonoSDF (Grid) Yu et al. (2022b) | 90.93 | 3.23 | 85.91 |
| DebSDF (MLP) Yu et al. (2022b) | **93.23** | 2.90 | 88.36 |
| Ours | 92.85 | **2.57** | **91.60** |

Table 4: **Quantitative results of Replica dataset**. Note that DebSDF reported results only with MLP representations.

| Method | Omnidata Eftekhar et al. (2021) | Wizard Fu et al. (2024) |
|---|---|---|
| Base | 0.700 | 0.661 |
| Ours | **0.750** | **0.730** |

Table 5: **F-score of Different Priors Models on ScanNet++.**

score compared to previous methods. Additional comparison results can be found in Appendix C.3, where our method recovered substantial detailed structures.

**ScanNet++:** ScanNet++ comprises numerous complex indoor scenes, each offering high-resolution GT mesh and RGB data. Since no public results are available for ScanNet++, we trained several baselines methods, including VolSDF, MonoSDF, DebSDF and Baked-Angelo Yu et al. (2022a). Among these, MonoSDF and DebSDF utilized monocular priors as we did. The quantitative results, summarized in Table 3, demonstrate the state-of-the-art performance of our method, particularly in terms of the Recall metric. This highlights the capability of our approach to recover significantly finer structures of the GT surface. Additional visualization results are prensented in Appendix C.2.

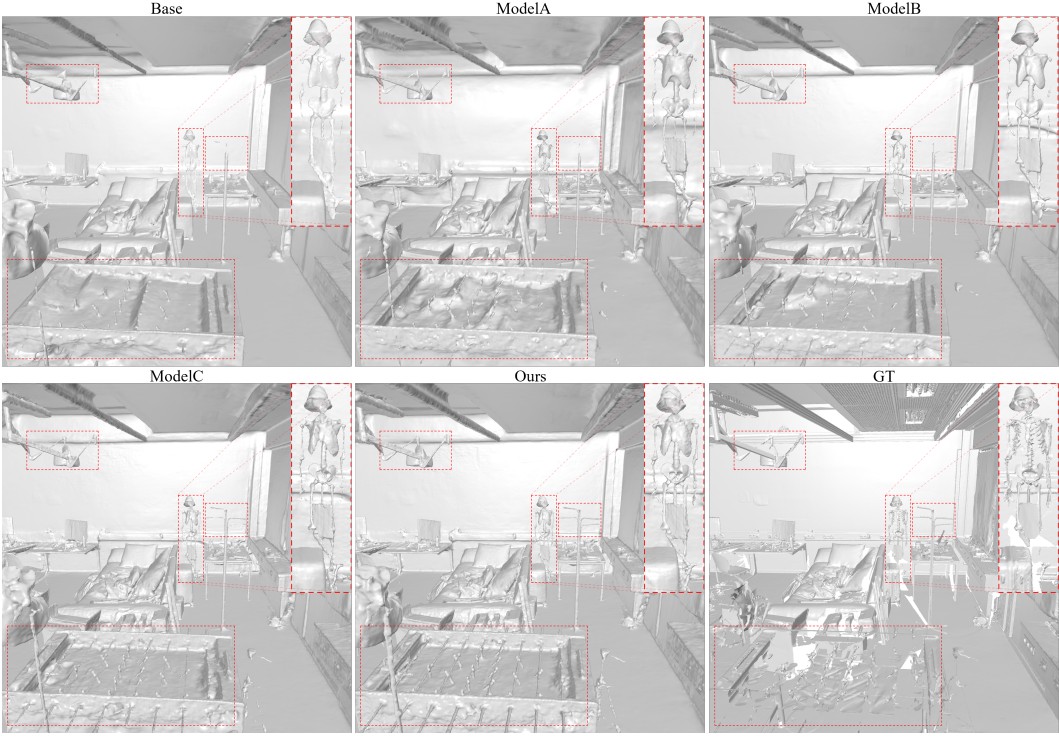

Figure 5: **Visualization Results of the Models after Ablation on ScanNet++.** Detailed definitions of these models and corresponding quantitative results are provided in Table 6.

## 4.2 ABLATION STUDY

### 4.2.1 ABLATION OF DIFFERENT MODULES

The learned deviations play a crucial role in locating high-frequency regions. Consequently, we employ deflection angles for sampling, photometric optimization, and unbiased rendering to facilitate the recovery of thin and fine-grained structures. We conducted detailed ablation experiments to assess the effectiveness of these modules, resulting in five post-ablation models: Base, ModelA,

| Method | Base | ND | AP | DO | DU | — | Acc↓ | Comp↓ | Prec↑ | Recall↑ | Chamfer↓ | F-score↑ |
|--------|------|-----|-----|-----|-----|---|------|-------|-------|---------|----------|----------|
| Base | ✓ | × | × | × | × | — | 0.046 | 0.046 | 0.576 | 0.640 | 0.046 | 0.606 |
| ModelA | ✓ | ✓ | × | × | × | — | 0.044 | 0.042 | 0.601 | 0.666 | 0.043 | 0.632 |
| ModelB | ✓ | ✓ | ✓ | × | × | — | 0.038 | 0.032 | 0.637 | 0.726 | 0.035 | 0.679 |
| ModelC | ✓ | ✓ | ✓ | ✓ | × | — | 0.040 | **0.030** | 0.631 | **0.740** | 0.035 | 0.681 |
| Ours | ✓ | ✓ | ✓ | ✓ | ✓ | — | **0.038** | 0.031 | **0.648** | 0.728 | **0.034** | **0.686** |

Table 6: **Quantitative Results of Ablation for Different Modules on ScanNet++.** 'Base' can be viewed as a simple combination of MonoSDF and Neuralangelo. 'ND' denotes applying the deflection field and the deflected normal loss term $L_{normal}^d$. 'AP' refers to the adaptive deflection angle prior loss terms including $L_{normal}^{ad}$ and $L_{depth}^{ad}$. 'DO' denotes the two deflection angle guided optimizations, namely sampling and photometric loss. 'DU' denotes the deflection angle guided unbiased rendering.

ModelB, ModelC, and Ours. For quantitative results and comprehensive definitions, please refer to Table 6. Figure 5 visually illustrates how the proposed components enhance overall performance.

**Analysis of Normal Deflection Field** The Base model strictly adheres to monocular cues, leading to significant detail loss due to misguided priors in complex areas. In ModelA, we introduced the deflection field and directly applied the deflected normal loss to learn deviations. The improvement in the F-score to 0.632 (as shown in Table 6) confirms the superiority of the deviation learning approach. However, we observed wrinkles (as depicted in Figure 5) in textureless regions like walls. This issue arises from the lack of constraints on the deflection, resulting in inefficient utilization of priors in smooth regions. In Model B, the introduction of the adaptive deflection angle prior loss significantly enhances reconstruction quality, striking a balance between smoothness and detail.

**Analysis of deflection angle guided optimization** While ModelB substantially improves reconstruction quality compared to the Base, it still struggles with finer structures (as depicted in Figure 5). In ModelC, we introduced deflection angle guided sampling and color loss to emphasize surface details. Finally, we incorporated an adaptive unbiased rendering method to facilitate the recovery of thin structures. Figure 5 illustrates that Ours recovers a wide range of complex and fine structures, such as skull models and iron swabs, surpassing the Base model. As indicated in Table 6, Ours achieved the highest F-score, quantitatively outperforming other models. This ablation study strongly validates the effectiveness of the proposed modules in restoring high-fidelity surfaces.

### 4.2.2 ABLATION OF DIFFERENT PRIOR MODELS

Our method learns deviations between the scene and normal priors predicted by a specific pretrained model. It is essential to evaluate its effectiveness across cues generated by different pretrained models. To achieve this, we employed the state-of-the-art method Geowizard Fu et al. (2024) to generate both depth and normal cues. The quantitative results, summarized in Table 5, demonstrate that our method significantly enhances surface quality irrespective of the cues utilized. This experiment illustrates the superiority of our approach. It invariably improves reconstruction accuracy in scenarios where prior knowledge is used.

## 5 CONCLUSION

We have presented ND-SDF, a novel approach that learns deviations between the scene and normal priors for high-fidelity indoor surface reconstruction. We introduce an adaptive deflection angle prior loss to dynamically supervise areas with varying characteristics. By identifying high-frequency regions based on deflection angles, we employ angle guided optimization to generate thin and fine-grained structures. Our method recovers a substantial number of complex structures, as demonstrated by extensive qualitative and quantitative results that confirm its superiority. ND-SDF efficiently adapts to unreliable priors in challenging areas, significantly addressing the detail loss issue caused by large prior errors in regions with fine structures.

**Limitation** ND-SDF is confined to reconstruction scenarios involving priors. Our approach exhibits limited capability in handling areas with less coverage. Due to inherent ambiguities in observations, the deflection field may learn incorrect structures and converge to local optima. We encourage the reader to refer the supplementary material for more analysis and results. In the future, we may explore multi-view consistency constraints to enhance reconstruction quality.

ACKNOWLEDGMENTS

This work was partially supported by Key R&D Program of Zhejiang Province (No.2023C01039) and NSF of China (No.62425209).

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

# APPENDIX / SUPPLEMENTARY MATERIAL

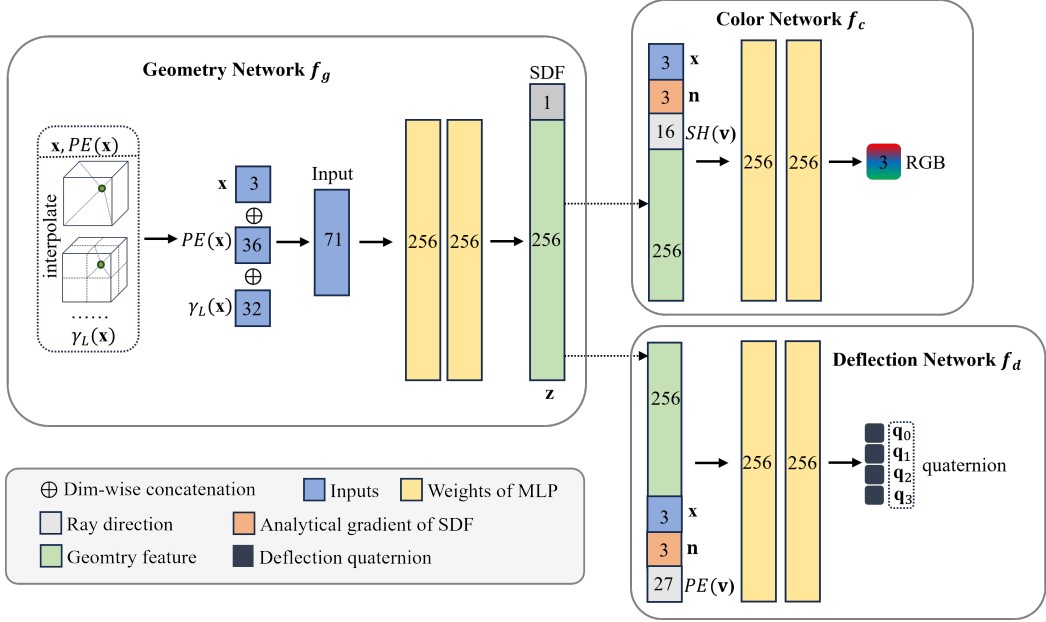

Figure 6: **Architecture of ND-SDF.** We utlize multi-resolution hash grids $\gamma_L$ for scene representation. The geometry network $f_g$, color network $f_c$, and deflection network $f_d$ are all constructed using simple multilayer perceptrons (MLPs).

## A   IMPLEMENTATION DETAILS

### A.1   ARCHITECTURE

When considering the inputs to the deflection network, two scenarios are contemplated: (1) No deflection is required when the prior is completely accurate. In this case, the quaternion axis aligns with the scene normals, indicating that the deflection axis is closely related to the scene geometry; (2) Since the prior model is view-dependent, the quaternion for alignment with the prior must also be view-dependent. Consequently, we define:

$$\mathbf{q} = f_d(\mathbf{x}, \mathbf{v}, \mathbf{n}, \mathbf{z}), \tag{17}$$

where $\mathbf{q} = (\mathbf{q}_0, \mathbf{q}_1, \mathbf{q}_2, \mathbf{q}_3)$ represents the deflection quaternion. As detailed in Section 3.2, the quaternion can be expressed in trigonometric form as $\mathbf{q} = \cos(\theta/2) + \sin(\theta/2)\left(\mathbf{u}^1 i + \mathbf{u}^2 j + \mathbf{u}^3 k\right)$. Within the deflection network, the rotation axis is initialized along the x-axis, i.e., $\mathbf{u} = (1, 0, 0)$, and the rotation angle $\theta/2$ is set to $\pi/2$. By default, the network normalizes the output quaternions.

We observe that the structure of the deflection network mirrors that of the color network Yariv et al. (2020), denoted as $\mathbf{c} = f_c(\mathbf{x}, \mathbf{v}, \mathbf{n}, \mathbf{z})$. However, considering that the color network exclusively models scene-related radiance, while the deflection network encapsulates a broader range of prior deviations, we separate these two networks.

### A.2   PROGRESSIVE WARM-UP

In the early stages of training, deflection fields may introduce negative effects, such as noise. Before the rough formation of scene geometry, these deflection quaternions are random and erroneous. If unconstrained, deflecting the true normals during this initial phase can lead to reconstruction errors, such as surface protrusions.

To address this issue, we propose a gradual activation strategy for deflection fields, allowing them to become effective as the surface takes shape. Specifically, we design a "quat anneal" policy to

| Method | w/o warm-up | w warm-up |
|--------|-------------|-----------|
| F-score | 0.667 | **0.686** |

Table 7: **Ablation of Warm-Up Strategy on ScanNet++.** The warm-up strategy can better facilitate the initialization of the deflection field, resulting in improved surface reconstruction quality.

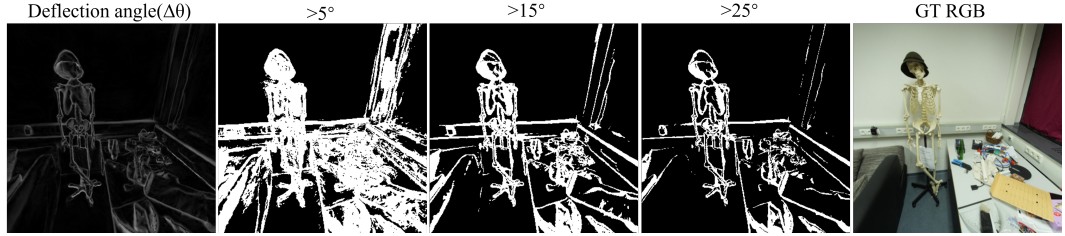

Figure 7: **Visualization of the Deflection Angle.** We visualize the angle map and the angle mask using thresholds of 5°, 15°, and 25°.

enable the deflection effect to progress gradually. Given the surface normals synthesized by volume rendering, denoted as $\hat{\mathbf{N}}(\mathbf{r})$, and the synthesized quaternion $\mathbf{Q}(\mathbf{r}) = (\mathbf{Q}_0, \mathbf{Q}_1, \mathbf{Q}_2, \mathbf{Q}_3)$, we first decompose $\mathbf{Q}(\mathbf{r})$ into its trigonometric representation:

$$\begin{cases} \dfrac{\theta}{2} = \arccos \mathbf{Q}_0 \\ \mathbf{u} = \dfrac{[\mathbf{Q}_1, \mathbf{Q}_2, \mathbf{Q}_3]}{\sin \frac{\theta}{2}} \end{cases}. \tag{18}$$

Here, $\mathbf{u}$ denotes the rotation axis corresponding to the quaternion $\mathbf{Q}(\mathbf{r})$, and $\theta$ represents the rotation angle. We then utilize the scene normals to warm up the deflection axis, starting with zero rotation:

$$\begin{cases} \dfrac{\theta_{iter}}{2} = prog_q \times \dfrac{\theta}{2} \\ \mathbf{u}_{iter} = prog_q \times \mathbf{u} + (1 - prog_q) \times \hat{\mathbf{N}}(\mathbf{r}) \end{cases}. \tag{19}$$

Here, the parameter $prog_q$ linearly increases from 0 to 1 during the training process within the process interval $[0, anneal\_quat\_end]$. Once the training progress reaches $anneal\_quat\_end$, the warm-up phase concludes, and $prog_q$ becomes equal to 1.

The warm-up deflection quaternion is computed as:

$$\mathbf{Q}_{iter}(\mathbf{r}) = cos\dfrac{\theta_{iter}}{2} + sin\dfrac{\theta_{iter}}{2} \times \mathbf{u}_{iter}. \tag{20}$$

During the initial training stage, the deflection angles start at zero and gradually increase, allowing the model to effectively leverage prior knowledge to optimize smooth regions and approximate the scene geometry. We consider it to be adequately initialized upon the warm-up phase concludes. Throughout this process, the deflection field gradually adapts to more complex structures and learns the deviations between the true geometry and the normal priors. Quantitative results with and without progressive warm-up are presented in Table 7.

## A.3 DETAILED IMPLEMENTATION ASPECTS

### A.3.1 ADAPTIVE DEFLECTION ANGLE PRIOR LOSS

In Section 3.3, we introduce two modulation functions, namely $g^d$ and $g$ (Eq. 12), which adjust the weights of the deflected and original normal loss terms based on the deflection angle.

Here provide the specific forms for both functions. We employ a shifted logistic function to define them:

$$\begin{cases} g^d(\theta) = \dfrac{1}{1 + e^{-s_0}(\theta - \theta_0)}. \\ g(\theta) = 1 - g^d(\theta) \end{cases} \tag{21}$$

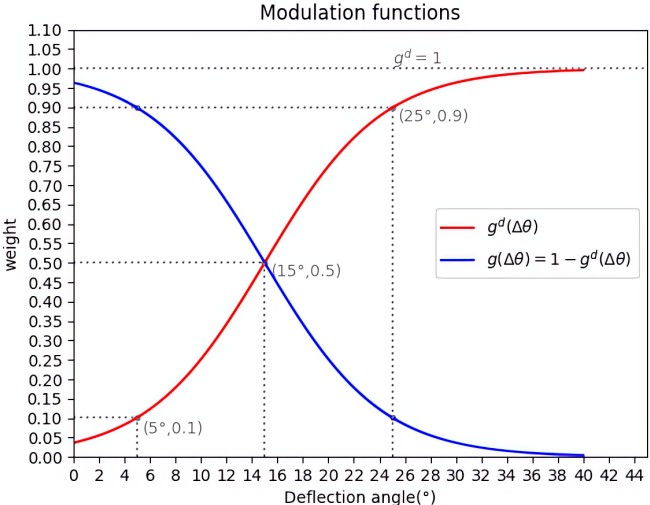

Figure 8: **Modulation Functions defined in Eq. 21.**

Here, the parameter $s_0$ controls the steepness of $g^d$, and $\theta_0$ denotes the offset term. In our experiments, we set $s_0 = 12.5, \theta_0 = \frac{\pi}{12}$ for all scenes.

As shown in Figure 7, a deviation angle of less than 5° consistently indicates a simple flat region, whereas an angle greater than 15° typically denotes a complex structural region.

Additionally, we visualize the modulation function $g^d(\Delta\theta)$ in Figure 8, with $(s_0, \theta_0)$ set to $(12.5, \frac{\pi}{12})$. For regions with a deviation angle less than 5°, weights greater than 0.9 are assigned to the original normal loss term, ensuring the effective incorporation of normal priors in smooth regions.

To further demonstrate the effectiveness of the proposed adaptive deflection angle prior loss, we visualize three normal loss heatmaps: the naive normal loss, the deflected normal loss, and the adaptive deflection angle normal loss (as shown in Figure 9). The black box highlights thin structures and ambiguous regions where monocular priors are largely unreliable. In these areas, the naive normal loss heatmap shows significant errors, while both the deflected and adaptive losses exhibit significantly lower errors by aligning the deflected scene normals with the normal priors (GT). In summary, the adaptive prior loss can dynamically adjust the weighting between the naive and deflected loss terms, promoting globally correct convergence while fully leveraging the normal prior.

### A.3.2 Deflection Angle Guided Optimization

In Section 3.4, we introduce three deflection angle guided optimization methods, including ray sampling, photometric optimization, and unbiased rendering.

To implement ray sampling guided by the deflection angle, we dynamically maintain a deflection angle map $(\Delta\bar{\theta})$ for each image during the training process. We update the angle map using the following equation:

$$\Delta\bar{\theta}(\mathbf{r}) = \max\left(\Delta\bar{\theta}^{old}(\mathbf{r}) \times \eta, \Delta\theta(r)\right), \tag{22}$$

where $\mathbf{r}$ is the sampled ray and $\eta$ is the decay coefficient. We initialize the angle map to zero at the beginning.

In the sampling phase, we first calculate the per-pixel sampling probability based on the angle map, then use this probability map for inverse sampling, such that areas with greater deflection magnitude tend to have more rays sampled.

We employ a scaled and shifted logistic function to calculate the per-pixel sampling probability:

$$p(\mathbf{r}_i) = 1 + \frac{t_1}{1 + e^{-s_1(\Delta\bar{\theta}(\mathbf{r}_i) - \theta_1)}}, \tag{23}$$

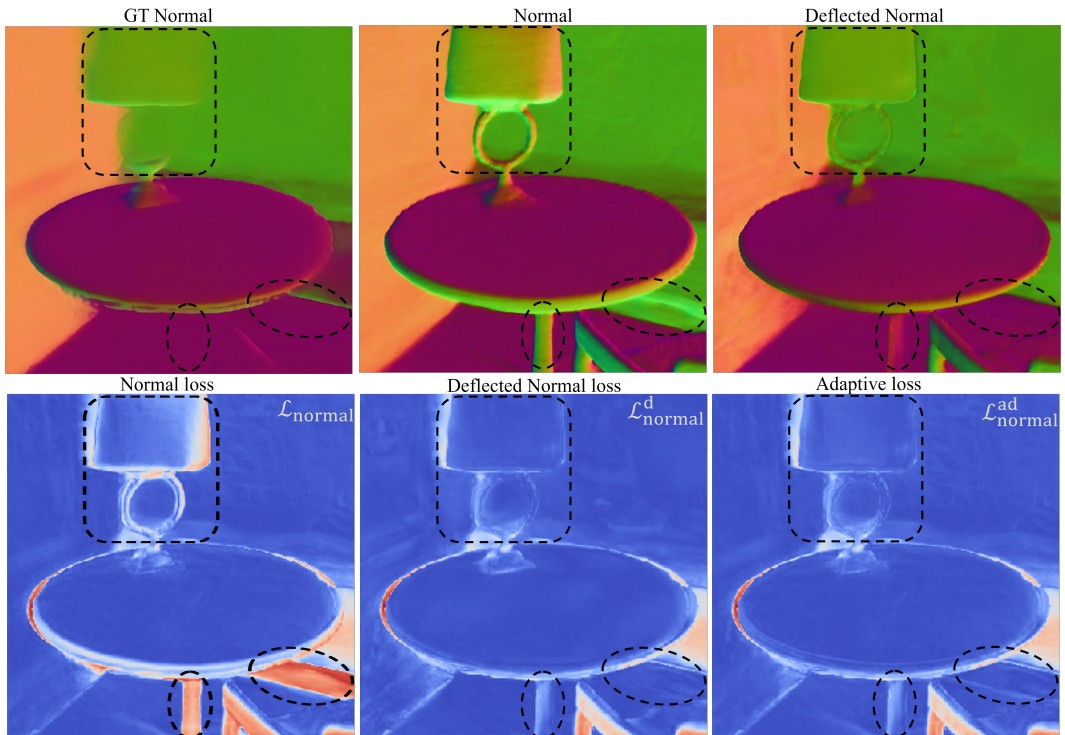

Figure 9: **Heatmap comparison of naive, deflected, and adaptive deflection angle normal loss.**

where $\Delta\bar{\theta}$ represents the maintained angle map, and $p(\mathbf{r}_i) \in [1, t_1+1]$, with the steepness controlled by $s_1$. We then normalize the probability map for inverse sampling using the following equation:

$$p(\mathbf{r}_i) = \frac{p(\mathbf{r}_i)}{\sum_i p(\mathbf{r}_i)}. \tag{24}$$

For the angle guided color loss, we similarly employ a scaled and shifted logistic function to re-weight the original color loss term based on the deflection angle (Eq. 14):

$$w_{color}(\Delta\theta(\mathbf{r})) = 1 + \frac{t_2}{1 + e^{-s_2(\Delta\theta(\mathbf{r}) - \theta_2)}}. \tag{25}$$

Regarding the unbiased rendering guided by the deflection angle, we first define a bias confidence using another variant of the logistic function to compute it:

$$cfd(\mathbf{r}) = \frac{1}{1 + e^{-s_3(\Delta\bar{\theta}(\mathbf{r}) - \theta_3)}}. \tag{26}$$

Note that we utilize the maintained angle map to calculate the confidence instead of the deflection angle $\Delta\theta(\mathbf{r})$ computed from current iteration, as unbiased densities of the sampled points along the ray are required prior to the volume compositing process. This then leads to the deflection angle guided unbiasing:

$$\sigma(\mathbf{r}(t_i)) = \frac{1}{\beta}\Psi_\beta\left(\frac{-f_g(\mathbf{r}(t_i))}{cfd(\mathbf{r})\left|f_g'(\mathbf{r}(t_i))\right| + 1 - cfd(\mathbf{r})}\right). \tag{27}$$

In practice, we set the parameters as follows: $(s_1 = 25, \theta_1 = \frac{\pi}{12}, t_1 = 4)$, $(s_2 = 25, \theta_2 = \frac{\pi}{12}, t_2 = 2)$, $(s_3 = 25, \theta_3 = \frac{\pi}{18})$.

### A.3.3 NUMERICAL GRADIENT

Following Neuralangelo Li et al. (2023b), we employ numerical gradients w.r.t multi-resolution hash grids to promote surface consistency, as proved effective in their work. The numerical gradient is computed as follows (considering only the SDF value output of geometry network $f_g$):

$$\nabla_x f_g(\mathbf{x}_i) = \frac{f_g(\gamma_L(\mathbf{x}_i + \boldsymbol{\epsilon}_x)) - f_g(\gamma_L(\mathbf{x}_i - \boldsymbol{\epsilon}_x))}{2\epsilon}. \tag{28}$$

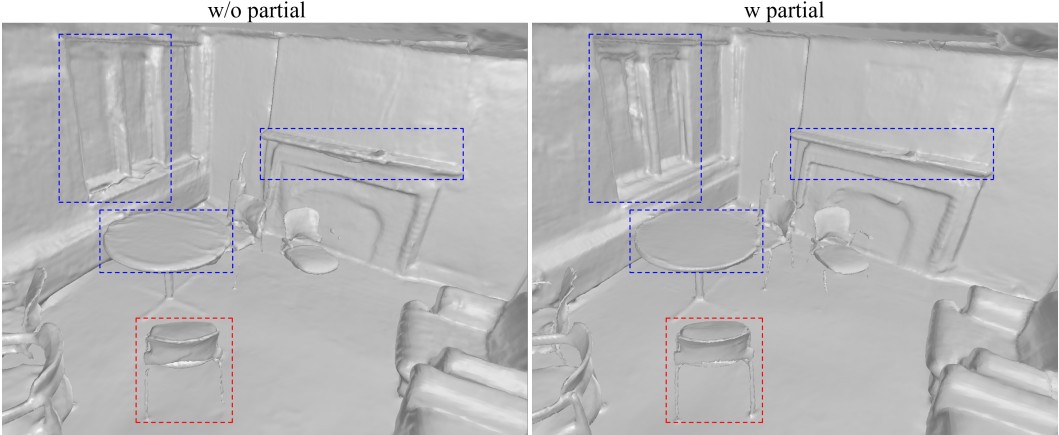

Figure 10: **Visual comparisons of partial unbiased rendering on ScanNet**.

Here, $\epsilon_x = [\epsilon, 0, 0]$ signifies the incremental step size along the x-axis. The method for calculating the numerical gradient along the y or z-axis remains the same.

We also utilize the curvature loss Li et al. (2023b) to further encourage the smoothness of the reconstructed surfaces. This loss term is defined as:

$$\mathcal{L}_{\text{curv}} = \frac{1}{N} \sum_{i=1}^{N} \left| \nabla^2 f_g \left( \mathbf{x}_i \right) \right|. \tag{29}$$

This term decreases exponentially during training, similar to Neuralangelo. In fact, our tests indicate that it has minimal impact on surface smoothness, resulting in only a slight improvement compared to its absence. We attribute this to the use of monocular cues, as the cues are particularly accurate in smooth regions.

## B    Extensive ablation studies

### B.1    Ablation of partial unbiased rendering

In Section 3.5, we introduced an unbiased function. We partially apply this technique to thin structures indicated by the learned deflection angles, without doing so may result in an inability to converge and a decrease in reconstruction quality.

We attribute this issue to the inconsistency in perspective inherent in the transform function. We compute the derivative of $f_g$ (considering only its SDF value output) w.r.t the sampling distance $t_i$ as follows:

$$f_g'(\mathbf{r}(t_i)) = \nabla f_g(\mathbf{o} + t_i \mathbf{v}) \cdot \mathbf{v} = \mathbf{n}_i \cdot \mathbf{v}, \tag{30}$$

where $\mathbf{v}$ is the ray direction and $\mathbf{n}_i$ represents the geometry normal at point $\mathbf{x}_i$. It means that $\frac{\partial f_g(\mathbf{r}(t_i))}{\partial t_i}$ is equivalent to the cosine of the ray direction and the normal.

Consider another ray in space, denoted as $\mathbf{r}'(t) = \mathbf{o}' + t\mathbf{v}'$. Assuming that this ray also passes through $\mathbf{x}_i$, sampling the same point such that $\mathbf{x}_j = \mathbf{o}' + t_j \mathbf{v}' = \mathbf{x}_i$. According to Eq. 15, the volumetric densities of $\mathbf{x}_i$ and $\mathbf{x}_j$ under the transform function are given by:

$$\sigma(\mathbf{r}(t_i)) = \frac{1}{\beta} \Psi_\beta \left( \frac{-f_g(\mathbf{r}(t_i))}{\left| f_g'(\mathbf{r}(t_i)) \right|} \right), \sigma(\mathbf{r}'(t_j)) = \frac{1}{\beta} \Psi_\beta \left( \frac{-f_g(\mathbf{r}'(t_j))}{\left| f_g'(\mathbf{r}'(t_j)) \right|} \right). \tag{31}$$

We observe that $f_g(\mathbf{r}(t_i)) = f_g(\mathbf{r}'(t_j))$ because $\mathbf{x}_i = \mathbf{x}_j$, but $\left| f_g'(\mathbf{r}(t_i)) \right| \neq \left| f_g'(\mathbf{r}'(t_j)) \right|$ because $\mathbf{n}_i \cdot \mathbf{v} \neq \mathbf{n}_j \cdot \mathbf{v}'$ even though $\mathbf{n}_i = \mathbf{n}_j$. The inconsistency in perspective leads to ambiguity in the volume density at the same sampling point, causing complex indoor surfaces to fail to converge. Therefore, to reduce this ambiguity as much as possible, we choose to apply the transform function only to fine regions indicated by the deflection angle.

| Method | w/o partial | w partial |
|---|---|---|
| F-score | 0.707 | **0.746** |

Table 8: **Ablation of Partial Unbiased Rendering on ScanNet.** We observed a decrease in reconstruction quality after omitting partial unbiased rendering.

| Method | w/o $\mathcal{L}_{depth}^{d}$ | w $\mathcal{L}_{depth}^{d}$ |
|---|---|---|
| F-score | 0.664 | **0.686** |

Table 9: **Ablation of adaptive depth prior loss on ScanNet++.** We observed a decrease in reconstruction accuracy after omitting the adaptive depth loss.

The quantitative ablation results on ScanNet for partial unbiased rendering are presented in Table 8, where the improved F-score highlights the effectiveness of this strategy.

Figure 10 provides qualitative comparisons. As indicated by the blue box, the surfaces reconstructed without the proposed partial unbiased rendering exhibit wrinkles and irregularities, and the fine structural details of the chair legs (highlighted in red) are not well captured. With the application of this strategy, the surfaces become significantly smoother and more precise. These visual results underscore the advantages of the partial unbiased rendering approach.

We further visualize the inverse variance ($1/\beta$) of the SDF-density conversion function (Eq. 3) during training in Figure 11(a) and find that, without partial unbiasing, $1/\beta$ fails to converge to a relatively high value. In fact, the larger the inverse variance, the sharper and more accurate the SDF-density field becomes. Conversely, a smaller value indicates greater uncertainty in the geometric surface, which leads to lower reconstruction accuracy and failure to converge properly. As shown in Figure 11(b), an increase in $1/\beta$ results in the rendering weight becoming more concentrated near the surface. Without a partial strategy, the surface is learned inaccurately and appears blurry.

More discussions about unbiasing methods are presented in Appendix C.4

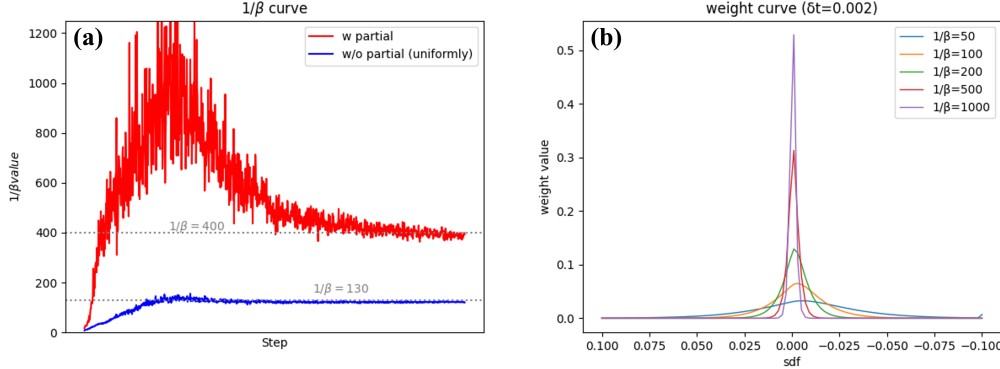

Figure 11: **(a)** The inverse variance curve of the SDF-density conversion function during training. **(b)** The rendering weight curve with respect to the inverse variance.

### B.2 ABLATION OF ADAPTIVE MONOCULAR DEPTH LOSS

Notably, we apply the same method to adjust depth supervision (Eq. 13) based on the deflection angle. This simplified treatment rests on a strong foundation for two reasons. First, both depth and normal cues share a common inductive bias, as they stem from the same pretrained model Eftekhar et al. (2021). Second, due to the inherent lack of scale in monocular depth, it cannot provide precise supervision. In implicit surface reconstruction with priors, depth cues are primarily utilized to facilitate convergence and accelerate surface formation. Therefore, automatically masking high-frequency areas based on the deflection angle has no adverse impact on the recovery of details.

We present the quantitative ablation results in Table 9. The improved performance underscores the importance of utilizing the adaptive deflection angle depth prior loss.

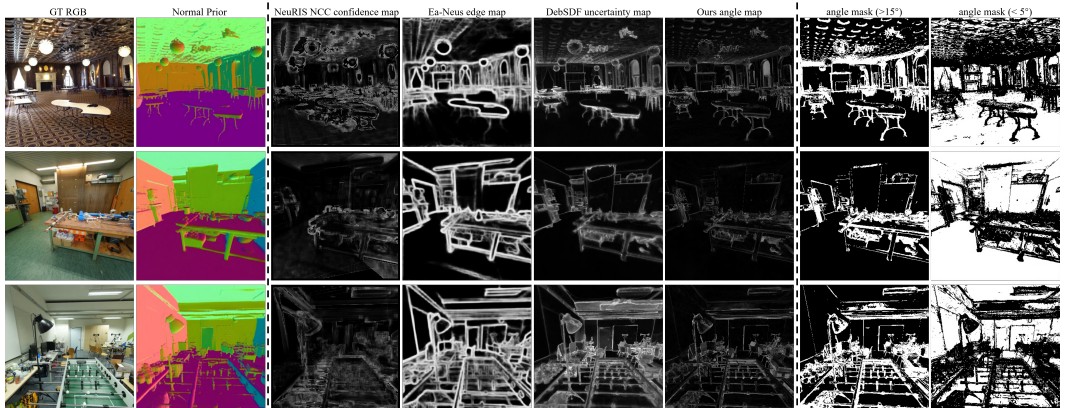

Figure 12: **Detailed visualization comparisons of structural region confidence maps obtained using various methods.** A comprehensive analysis is provided in Appendix B.3.

## B.3 CONFIDENCE MAP ANALYSIS

As illustrated in Figure 12, we present the confidence maps that highlight structural regions, which are obtained using a variety of methods, including the NCC map Wang et al. (2022), edge map Li et al. (2023a), uncertainty map Xiao et al. (2024) and our angle map. The NCC map exhibits low performance due to its simplistic local cross-correlation computing method, and it fails to properly distinguish high-frequency flat regions, such as textured floors. The edge map, generated by a pre-trained network Su et al. (2021), effectively captures object edges but still lacks precise localization of the fine structures. The uncertainty map, which is learned by constructing an uncertainty field, performs better than the first two methods. However, it only provides an approximate indication of high-frequency regions, as it inherently serves as an intensity field to measure the confidence of monocular cues, independent of actual geometry. As a result, the uncertainty map fails to precisely evaluate the actual differences between scene geometry and priors, leading to inefficient utilization of various priors. Additionally, it fails to recognize highlighted areas, as shown in the second row of Figure 12 and the first row of Figure 16. In contrast, our method, by modeling geometry-relevant SO(3) residual fields, accurately locates specific structural regions and provides precise angular deviations. The angle map clearly reflects the underlying geometry difference. Compared to the methods discussed above, our approach yields more reasonable and accurate results.

In terms of the sampling strategy guided by the confidence maps illustrated in Figure 12, our sampling process is fully weighted by geometry-aware angular deviations and is precisely localized to structural regions. In contrast, other methods, such as edge-aware and uncertainty-driven sampling, exhibit lower localization precision and are less sensitive to geometric differences from the priors, failing to provide effective sampling weights for various priors.

This analysis is further validated through extensive experiments. As shown in Figure 16 and Figure 17, our method generates more accurate fine structures compared to DebSDF.

## B.4 ANALYSIS OF THE ROBUSTNESS OF MONOCULAR CUES

To deeply investigate the effectiveness of our proposed explicit deflection-based method, we conducted experiments under the assumption that the priors for smooth regions are inaccurate. Specifically, we manually rotated all monocular normals in the smooth regions by angles ranging from 0° to 60°. A SAM model Medeiros et al. (2024) was employed to detect the flat areas. The rotation was performed by rotating the normal priors by a specified degree (e.g., 60°) toward a pre-defined axis. We evaluated MonoSDF, DebSDF, and ND-SDF on the 0e75f3c4d scene from ScanNet++. The results are summarized in Table 10 and Figure 13. As demonstrated in the results, decreases in F-score are observed across all methods. Specifically, MonoSDF produces an entirely abnormal result, as it rigidly relies on the biased priors. DebSDF also fails to maintain robustness in this scenario. Although the uncertainty field in DebSDF can easily mask irregular priors by learning uncertainty variances, it leads to inefficient utilization of the consistency information inherent in monocular cues within smooth regions. In contrast, our method explicitly models the actual bias amplitude,

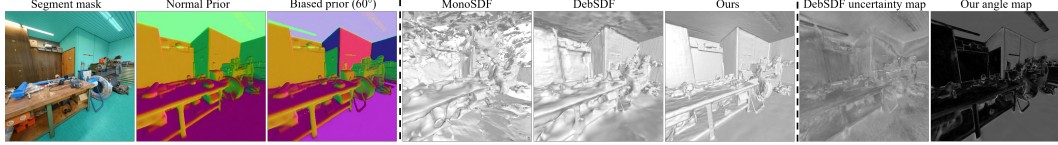

Figure 13: **Visualization results with monocular normal cues manually biased by 60 degrees.** Our method accurately captures the view-dependent biased priors and yields promising reconstruction result.

| Methods | 0° | 5° | 15° | 30° | 45° | 60° | 0°→60° |
|---------|------|------|------|------|------|------|--------|
| Ours | 0.750 | 0.735 | 0.678 | 0.654 | 0.709 | 0.655 | **-0.095** |
| MonoSDF | 0.633 | / | / | / | / | 0.140 | -0.493 |
| DebSDF | 0.701 | / | / | / | / | 0.517 | -0.184 |

Table 10: **Quantitative results under manually biased monocular normal cues.** The observed minimal decline in F-score underscores the effectiveness and robustness of our approach.

achieving notably robust performance with the smallest decrease in F-score (Table 10), even when the normal priors are biased by as much as 60°. These findings further validate the generalization and robustness of our method in non-trivial assumptions of the monocular cues.

## C  MORE RESULTS

### C.1  REPLICA

We present additional visualization results of our method on Replica in Figure 14.

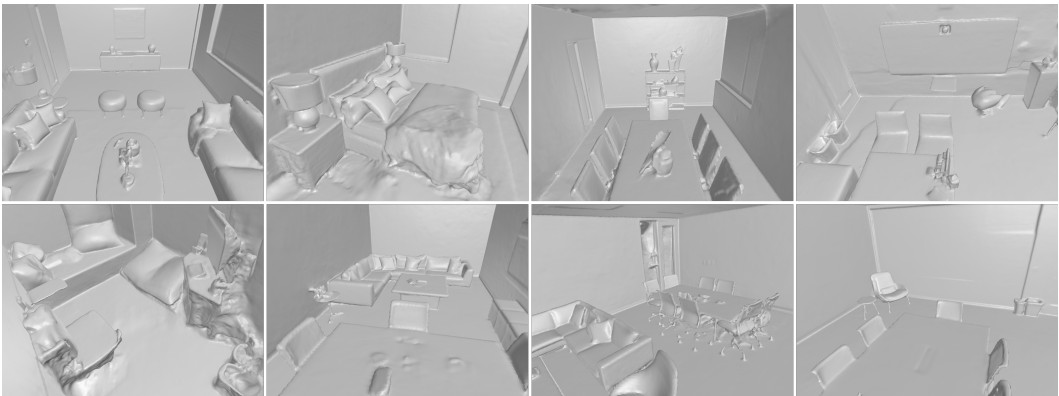

Figure 14: **Qualitative results of our method on Replica dataset.**

### C.2  SCANNET++

We present detailed quantitative results of the selected six scenes from ScanNet++ in Table 11 and provide more visual comparison results with previously state-of-the-art methods in Figure 15, Figure 16 and Figure 17. Compared to the baselines, our method yields significantly improved reconstructions both visually and quantitatively, notably excelling at capturing thin, fine-grained structures while maintaining smoothness in flat regions.

### C.3  TANKS AND TEMPLES

As illustrated in Figure 19 and Figure 20, we extracted the reconstructed meshes at a resolution of 2048. At this high resolution, the meshes demonstrate that our method significantly enhances the generation of small and thin structures, even in challenging large-scale indoor scenes. For all experiments, the hash grid settings were kept at their defaults, with resolutions ranging from $2^5$ to $2^{11}$ across 16 levels, and a hashmap size of $2^{19}$.

We present more qualitative comparisons with previous state-of-the-art baselines, including MonoSDF Yu et al. (2022b) and DebSDF Xiao et al. (2024) at 2048 resolution, as depicted in Figure 18. It is evident that ND-SDF achieves accurate dense surfaces replete with extensive details.

| Method | 036bce3393 | | | | | | 0e75f3c4d9 | | | | | |
|---|---|---|---|---|---|---|---|---|---|---|---|---|
| | Acc↓ | Comp↓ | Pre↑ | Recall↑ | Chamfer↓ | F-score↑ | Acc↓ | Comp↓ | Pre↑ | Recall↑ | Chamfer↓ | F-score↑ |
| VolSDF | 0.053 | 0.098 | 0.475 | 0.372 | 0.0755 | 0.417 | 0.053 | 0.078 | 0.395 | 0.363 | 0.065 | 0.378 |
| Baked-Angelo | 0.168 | 0.023 | 0.52 | 0.78 | 0.0955 | 0.624 | 0.121 | 0.053 | 0.525 | 0.719 | 0.087 | 0.607 |
| MonoSDF (Grid) | 0.037 | 0.037 | 0.606 | 0.656 | 0.037 | 0.630 | 0.064 | 0.027 | 0.574 | 0.705 | 0.046 | 0.633 |
| DebSDF (Grid) | 0.033 | 0.032 | 0.679 | 0.744 | 0.032 | 0.710 | 0.067 | 0.023 | 0.62 | 0.794 | 0.045 | 0.701 |
| ND-SDF | 0.037 | 0.031 | 0.648 | 0.728 | 0.034 | 0.686 | 0.054 | 0.018 | 0.662 | 0.864 | 0.036 | 0.750 |

| Method | 108ec0b806 | | | | | | 7f4d173c9c | | | | | |
|---|---|---|---|---|---|---|---|---|---|---|---|---|
| | Acc↓ | Comp↓ | Pre↑ | Recall↑ | Chamfer↓ | F-score↑ | Acc↓ | Comp↓ | Pre↑ | Recall↑ | Chamfer↓ | F-score↑ |
| VolSDF | 0.049 | 0.097 | 0.475 | 0.372 | 0.073 | 0.417 | 0.097 | 0.075 | 0.457 | 0.436 | 0.086 | 0.446 |
| Baked-Angelo | 0.092 | 0.050 | 0.498 | 0.624 | 0.071 | 0.554 | 0.420 | 0.024 | 0.457 | 0.799 | 0.222 | 0.582 |
| MonoSDF (Grid) | 0.041 | 0.045 | 0.575 | 0.576 | 0.043 | 0.576 | 0.138 | 0.022 | 0.708 | 0.813 | 0.080 | 0.757 |
| DebSDF (Grid) | 0.036 | 0.045 | 0.61 | 0.595 | 0.041 | 0.602 | 0.154 | 0.021 | 0.733 | 0.844 | 0.087 | 0.785 |
| ND-SDF | 0.047 | 0.040 | 0.591 | 0.646 | 0.044 | 0.617 | 0.112 | 0.015 | 0.721 | 0.888 | 0.064 | 0.796 |

| Method | ab11145646 | | | | | | e050c15a8d | | | | | |
|---|---|---|---|---|---|---|---|---|---|---|---|---|
| | Acc↓ | Comp↓ | Pre↑ | Recall↑ | Chamfer↓ | F-score↑ | Acc↓ | Comp↓ | Pre↑ | Recall↑ | Chamfer↓ | F-score↑ |
| VolSDF | 0.089 | 0.164 | 0.321 | 0.226 | 0.126 | 0.265 | 0.077 | 0.101 | 0.307 | 0.262 | 0.089 | 0.283 |
| Baked-Angelo | 0.076 | 0.038 | 0.575 | 0.701 | 0.057 | 0.632 | 0.034 | 0.043 | 0.685 | 0.687 | 0.038 | 0.686 |
| MonoSDF (Grid) | 0.036 | 0.035 | 0.695 | 0.700 | 0.036 | 0.697 | 0.027 | 0.024 | 0.747 | 0.770 | 0.026 | 0.758 |
| DebSDF (Grid) | 0.038 | 0.036 | 0.685 | 0.686 | 0.036 | 0.685 | 0.03 | 0.027 | 0.684 | 0.697 | 0.028 | 0.69 |
| ND-SDF | 0.054 | 0.024 | 0.657 | 0.786 | 0.039 | 0.716 | 0.036 | 0.020 | 0.723 | 0.798 | 0.028 | 0.759 |

Table 11: **Quantitative results of the 6 selected scenes from ScanNet++**. ND-SDF significantly outperforms previous state-of-the-art methods in most metrics, particularly in terms of Recall.

## C.4 ADDITIONAL DISCUSSIONS

**More information about dataset.** The qualitative results on ScanNet presented in the main content (as shown in Figure 4) demonstrate that our method captures fine details, such as the chair legs, which are even overlooked by the ground truth (GT). This highlights a known issue regarding the reliability of ScanNet GT meshes. Constructed in 2017, the ScanNet dataset is based on RGB-D data captured by an iPad with an additional structure sensor. The images suffer from relatively low resolution, motion blur, and lighting variations. Furthermore, the limitations of the bundle-fusion algorithm result in GT surfaces lacking fine structures, which can adversely affect evaluation performance in these areas. This underscores our preference for conducting quantitative ablation studies on ScanNet++, which provides more reliable evaluation results.

**Concurrent work.** We notice that the concurrent work NC-SDF Chen et al. (2024b) share analogous idea with us. We clarify that our method is entirely independent of NC-SDF. Although we similarly model an SO(3) residual field to enhance surface reconstruction, our design diverges substantially. NC-SDF employs a two-stage approach, yet struggles with noise in the second stage's explicit deflection. In contrast, our adaptive prior loss automatically assigns weights based on sample characteristics (Eq. 12), removing the need for staged training. Additionally, NC-SDF requires extra boundary information for optimization, reducing robustness and adding hyperparameters. While NC-SDF was tested on datasets with simple layouts like ScanNet and ICL-NUIM, we evaluated our method on more complex indoor scenes, including ScanNet++ and T&T. Extensive experiments confirm our method's superior generalization ability in challenging environments, an advantage not observed in NC-SDF.

**About unbiasing methods.** It's crucial to resolve bias issues in SDF-based surface reconstructions. We show the necessity to employ unbiasing transformation through an actual sampling process in Figure 3, supported by ablation results in Figure 5. Inherently, bias refers to the failure of the rendering weight to attain its maximum value at the zero-level set of SDF, thereby leading to incorrect rendering depth. This issue not only damages the surface quality but also severely affects the generation of fine structures. Many works have been proposed to solve this: The unbias assumption of Neus Wang et al. (2021) ensures the local maximum of the weight for a local plane. But the weight is severely reduced when the viewing direction gets closer to the tangent plane. TUVR Zhang et al. (2023) transforms the input SDF value to the actual depth from the tangent plane and extends the

extremal property of weight to any surface distribution. But TUVR still generates abnormal bi-modal shaped weight when passing close to an object if $1/\beta$ is not large enough. To address this, DebSDF Xiao et al. (2024) proposed using curvature radius to correct the SDF value, tremendously reducing the bias. However, these unbiasing methods, such as TUVR and DebSDF, face serve convergence issues, as the transformation must be perspective-dependent, causing inconsistency. We analyze this issue using TUVR as example in Appendix B.1. To mitigate this, we utilize explicit learned angular deviations to help apply the unbiasing transformation only to low-proportion fine-grained structures in the scene, which has proved effective in our method. Simpler solutions also exist, such as applying unbiasing only after the surface has sufficiently converged. However, another issue arises: the implicit SDF field is difficult to reshape, making it challenging to capture fine structures in later stages. There are still many issues to be addressed.

## D    SOCIETAL IMPACT

Our method contributes to high-fidelity indoor surface reconstruction. On the positive side, precise indoor surface reconstruction can revolutionize fields such as architecture, virtual reality, and robotics. It enables architects to visualize and optimize designs, enhances immersive experiences in virtual environments, and aids in navigation for autonomous robots. However, there are also challenges. Misuse of this technology could invade privacy, as detailed indoor reconstructions might reveal sensitive information. Striking a balance between innovation and responsible deployment will be crucial for maximizing the positive impact while minimizing potential harm.

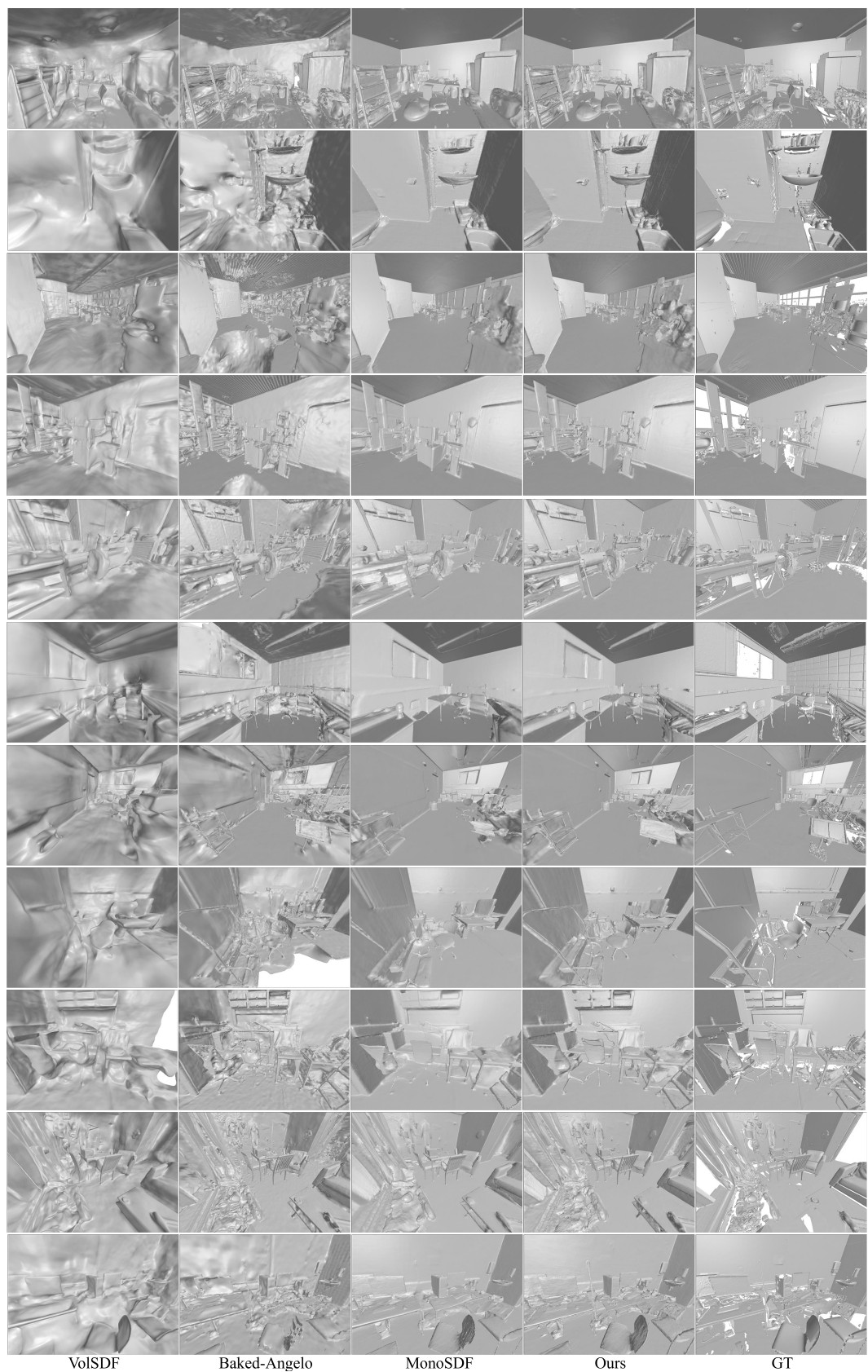

| VolSDF | Baked-Angelo | MonoSDF | Ours | GT |

Figure 15: **Qualitative Comparison on ScanNet++.** We compare ND-SDF with several baseline methods, including VolSDF, Baked-Angelo, and MonoSDF. Notably, only MonoSDF and Ours leverage monocular priors. Our method effectively balances smoothness and detail, whereas MonoSDF loses significant fine structures, and Baked-Angelo struggles to handle textureless smooth regions, such as floors.

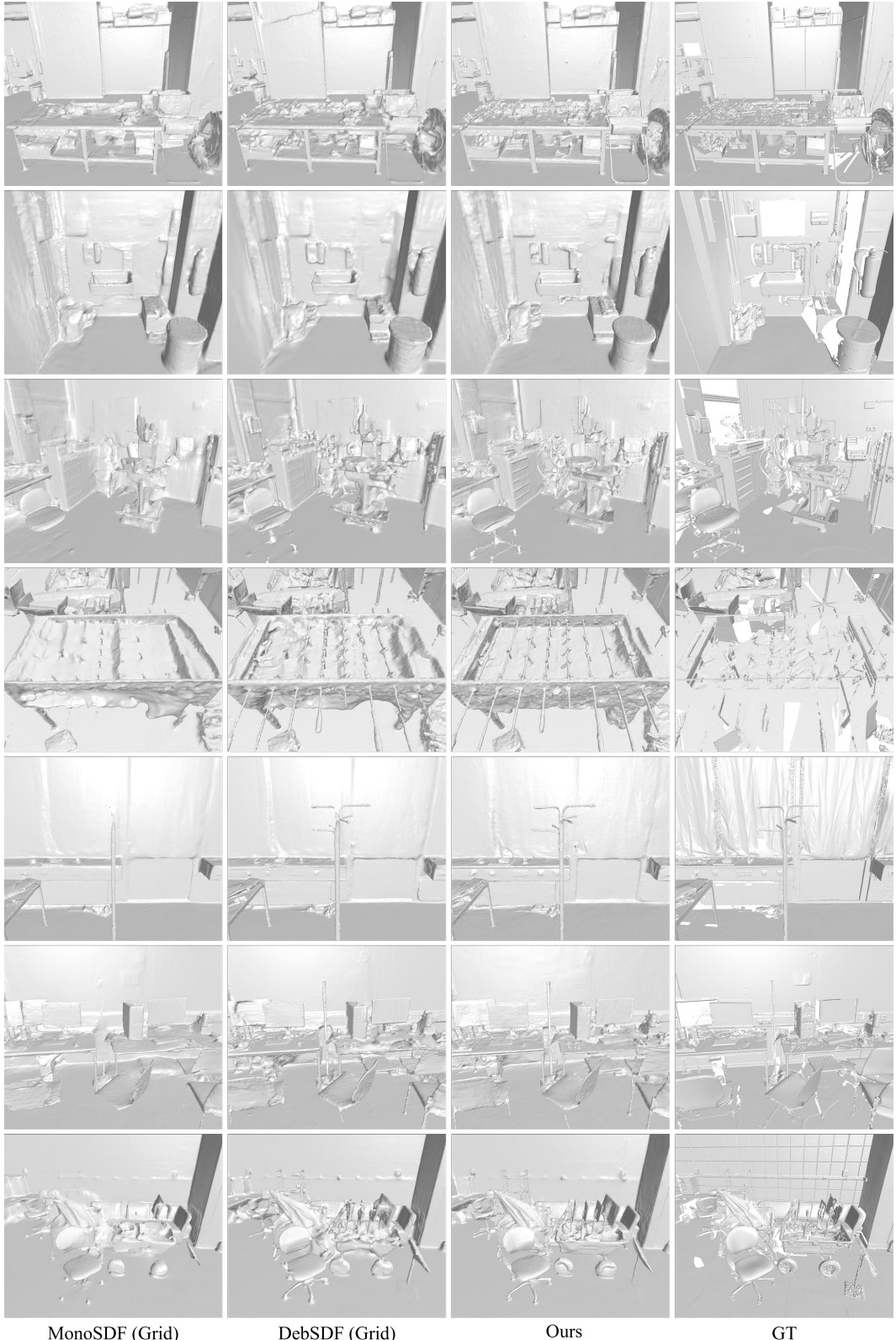

|  MonoSDF (Grid) | DebSDF (Grid) | Ours | GT |

Figure 16: **Additional Visualization Comparisons on ScanNet++.** We further compare our method with state-of-the-art method DebSDF. As shown above, ND-SDF tremendously outperforms previous methods when facing challenging complex indoor scenes. It shows a superior ability to capture accurate and geometry-consistent detailed structures.

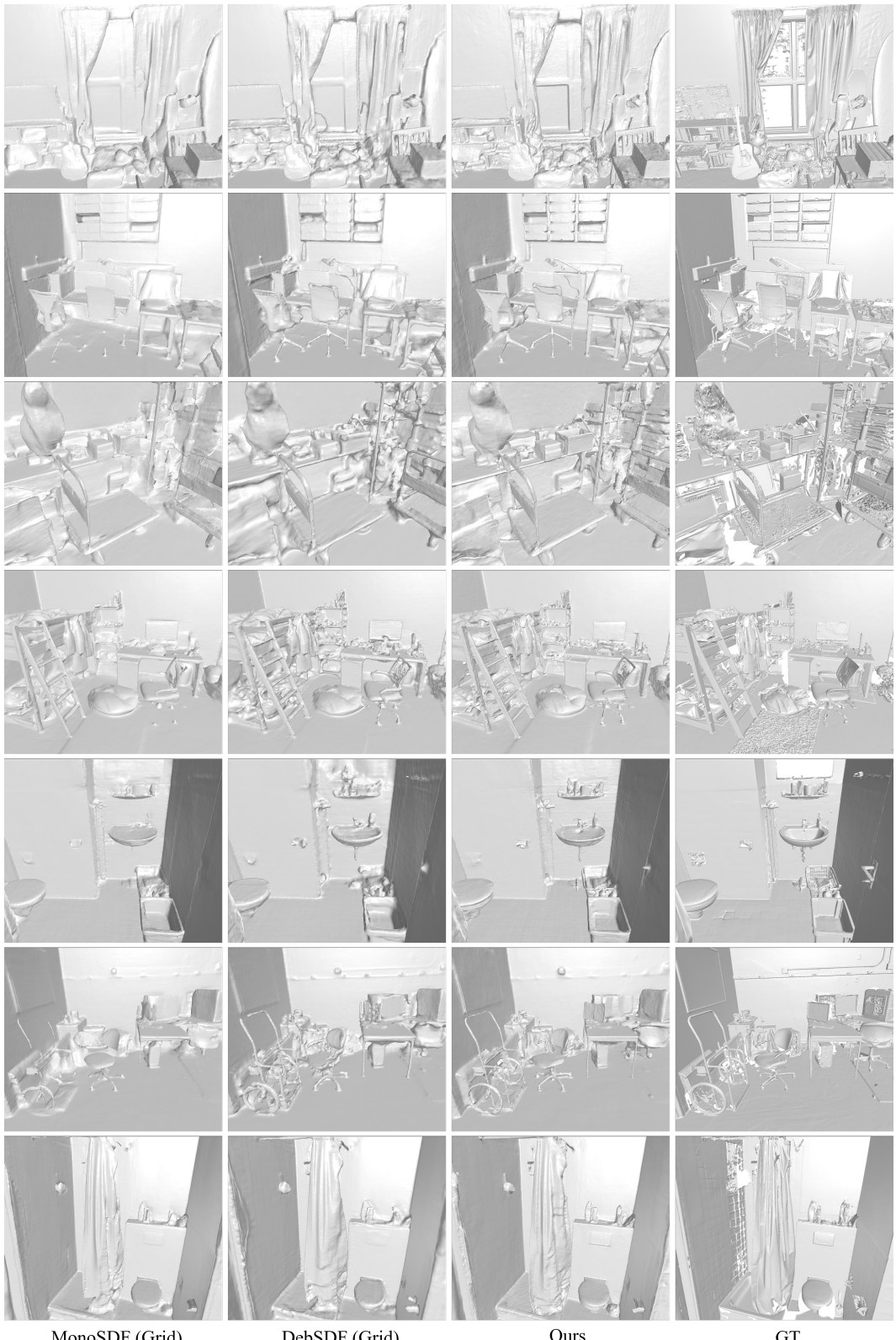

|  MonoSDF (Grid) | DebSDF (Grid) | Ours | GT |

Figure 17: **Additional Visualization Comparisons on ScanNet++.** We further compare our method with state-of-the-art method DebSDF. As shown above, ND-SDF tremendously outperforms previous methods when facing challenging complex indoor scenes. It shows a superior ability to capture accurate and geometry-consistent detailed structures.

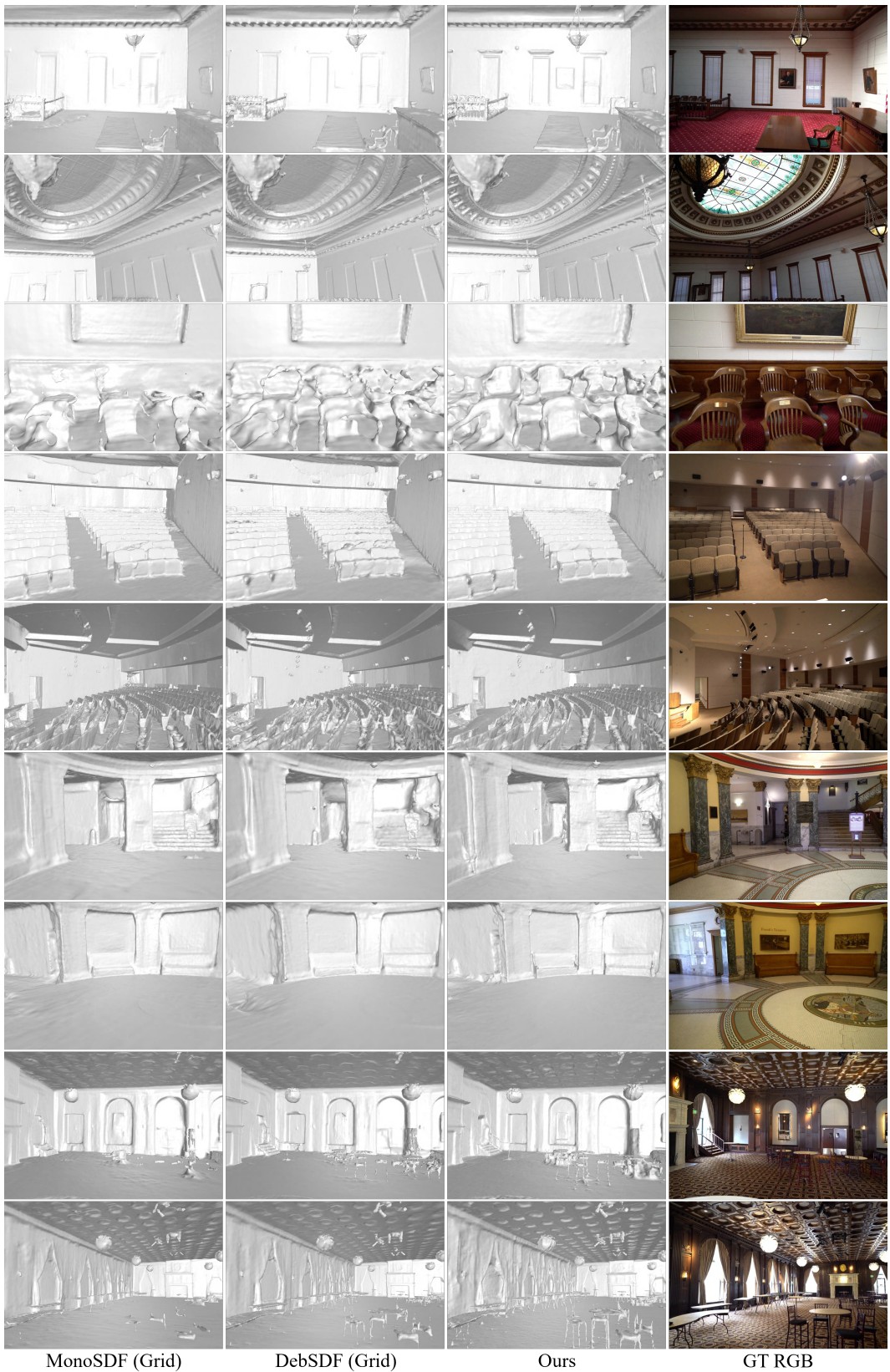

MonoSDF (Grid)          DebSDF (Grid)          Ours          GT RGB

Figure 18: **Qualitative Comparison on Tanks & Temples.** We compare ND-SDF with previous state-of-the-art indoor implicit reconstruction methods.

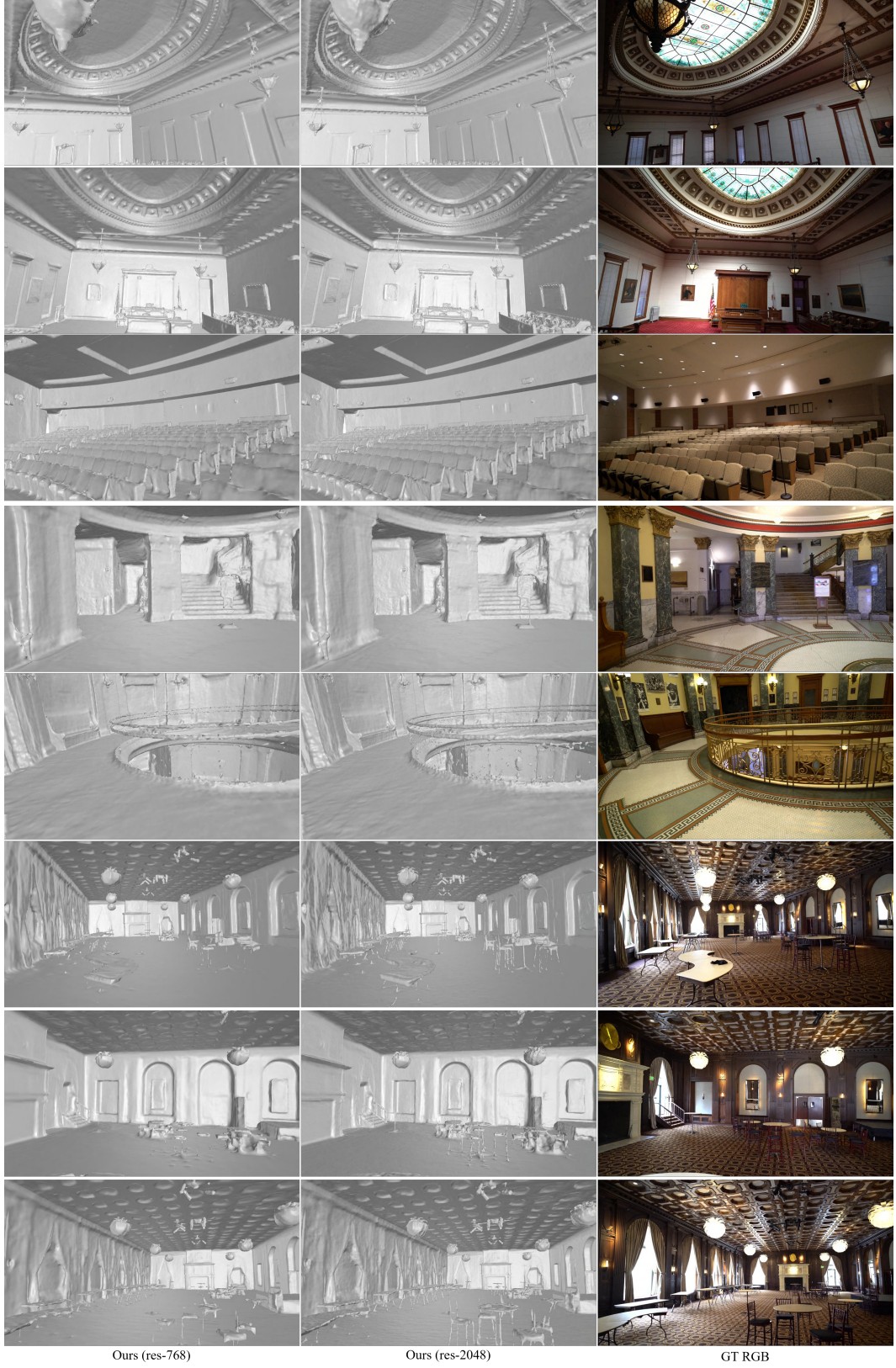

Ours (res-768)                Ours (res-2048)                GT RGB

Figure 19: **Qualitative Comparison of Different extraction resolutions on Tanks & Temples.**
We additionally extract the recovered mesh from the implicit SDF at a resolution of 2048.

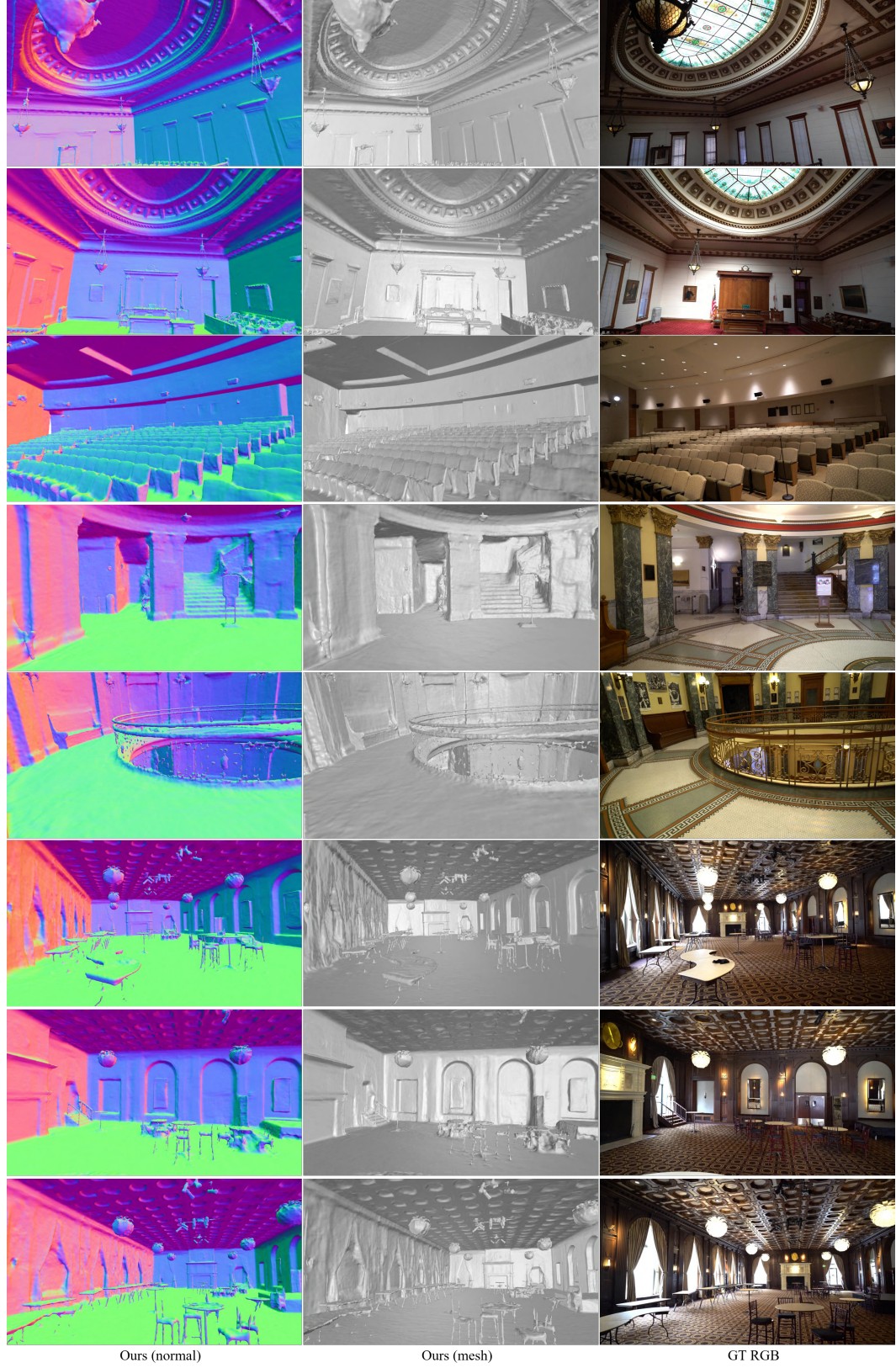

Ours (normal)        Ours (mesh)        GT RGB

Figure 20: **Qualitative Results of the extracted mesh at a resolution of 2048 on Tanks & Temples.**

