# OpenReview forum: "ND-SDF: Learning Normal Deflection Fields for High-Fidelity Indoor Reconstruction"
_ICLR.cc/2025/Conference — ICLR 2025 Spotlight_

### Official Review · Reviewer_M33k · 2024-10-27

**Soundness:** 2
**Presentation:** 3
**Contribution:** 3
**Rating:** 6
**Confidence:** 5

**Summary:**

This paper introduces ND-SDF and proposes to learn a Normal Deflection field to represent the angular deviation between the scene normal and the normal prior.

The paper also additionally introduces a ray sampling strategy based on the deflection angle to improve the surface quality further. The paper also proposes to use an unbiased function inspired by TUVR.

**Strengths:**

1. The paper writing is easy to follow.

2. The experiment covers lots of different dataset for a comprehensive comparison, like Scannet, TanksandTemples, Scannet++, Replica.

3. The ablation study indicates the effectiveness of the deflection field and adaptive angle prior when compared to its base model.

**Weaknesses:**

Major weakness:

1. Although the dataset included is various, the compared methods are not so much comprehensive(except the Scannet). The quantitative comparison on TanksandTemples, Scannet++, Replica only includes three works in 2022(NeuRIS, MonoSDF, VolSDF) and one work in 2021(Unisurf). This leads to an insufficient experiment. For example, what about the comparison with HelixSurf, TUVR, DebSDF, H2OSDF on TanksandTemples, Scannet++, Replica? The reviewer thinks the comparison without newer baselines is insufficient on TanksandTemples, Scannet++, Replica. Although Table 1 includes most newer baselines, usually only 4 scenes in Scannet are used for indoor scene reconstruction experiments. This is why the reviewer thinks that more quantitative experiments are required here.

Details: The reviewer understands that it's not necessary to run all previous baselines in all datasets. But just place here as an example. Since the unbiased weight technique is used, the reviewer thinks **at least** TUVR(CVPR2023) is a newer baseline that should be included in both the quantitative and qualitative comparisons on TanksandTemples, Scannet++, Replica datasets. (But more comparison with other newer methods will enhance the paper's experiment part.)

2. The qualitative comparison is also insufficient (only compared with older baselines). It's better to include the qualitative comparison with newer baselines in Figure 4, Figure 11, Figure 12, Figure 13.

3. Some works have both pure MLP version and hash grid version (like MonoSDF). The performance and efficiency of different versions have some differences. Since ND-SDF includes instant-NGP in its implementation, it's better to clearly indicate the version to compare within the table. (For example, "MonoSDF(mlp)", "MonoSDF(grid)".)

Minor weakness:

4. For Section 3.4, similar ideas of better ray sampling strategy have been investigated in some previous work. Like edge-aware sampling in EaNeuS[1], GaussianRoom[2], and uncertainty-aware sampling in DebSDF. Please consider adding a brief discussion about these previous methods.

5. In Figure 2, "GT depth" and "GT Normal" should be changed to "depth prior" and "normal prior".

6. The number of selected scenes in each dataset for the experiment has not been reported.

7. The dataset conducted in the ablation study has not been reported.

Reference:

[1] Edge-aware Neural Implicit Surface Reconstruction. Li et al. (ICME 2023)

[2] GaussianRoom: Improving 3D Gaussian Splatting with SDF Guidance and Monocular Cues for Indoor Scene Reconstruction. Xiang et al. (arxiv 2024)

**Questions:**

1. For Figure 3, what about the comparison with unbiased density function in NeuS, TUVR, DebSDF, HF-NeuS[3], and NeuRodin(optional, because NeuRodin is the concurrent work)[4]? These works also propose an unbiased density strategy. (visual comparison is better for good understanding)

2. In Figure 7, The deflection angle map appears to have a higher value at complex structural regions. What's the visual comparison between edge mask(EaNeuS), NCC mask(Equation 6 in NeuRIS), and uncertainty-aware mask(DDPNeRF[5], DebSDF, H2OSDF)?

3. For Equation 12 of this paper, it seems that when $\hat{\mathbf{N}}^d(\mathbf{r})$ and $\hat{\mathbf{N}}(\mathbf{r})$ perfectly fits the prior $\mathbf{N}(\mathbf{r})$, then this loss will become a perfect zero. The reviewer is concerned that this situation that causes zero loss doesn't seem to be in a good state. Is there a more comprehensive explanation about this training objective and why this training objective will work to help solve the concerns above?

4. Does the proposed normal deflection field have some advantages over the normal uncertainty field, like in DebSDF and H2OSDF?

Reference:

[3] HF-NeuS: Improved Surface Reconstruction Using High-Frequency Details. Wang et al. (Neurips 2022)

[4] NeuRodin: A Two-stage Framework for High-Fidelity Neural Surface Reconstruction. Wang et al. (Neurips 2024)

[5] Dense Depth Priors for Neural Radiance Fields from Sparse Input Views. Roessle et al. (CVPR 2022)

---

> ### Author Response · Authors · 2024-11-18
>
> We sincerely appreciate your time and effort in reviewing our work. Thank you for your thoughtful comments and constructive suggestions on our paper! Here, we address the concerns raised in the review:
> ## **Comparisons with newer methods**
> > Concerns about insufficient qualitative or quantitative comparison results with the latest methods, such as TUVR, and DebSDF.
>
> Additionally, we included DebSDF and TUVR for comparison. As TUVR is not open-sourced and was originally tested only on the DTU dataset, we incorporated TUVR's unbiasing function (Eq.15) into the official MonoSDF (Grid) framework to create the TUVR (MonoSDF-Grid) method, hereafter referred to simply as TUVR. The quantitative results are as follows:
>
> On Replica, we directly use the official evaluation results from DebSDF:
> |Method|TUVR|DebSDF (MLP)|Ours|
> |:-:|:-:|:-:|:-:|
> |F-score↑|85.81|88.36|**91.6**|
>
> ScanNet++:
> |Method|TUVR|DebSDF (Grid)|Ours|
> |:-:|:-:|:-:|:-:|
> |F-score↑|53.7|69.6|**72.1**|
>
> For the T&T dataset, to our knowledge, DebSDF was evaluated at 1024 resolution and MonoSDF at 768. To align with DebSDF, we increased the marching cubes resolution from 768 to 1024 and resubmitted it to the official T&T evaluation site. Additionally, we assessed results at a higher resolution of 2048, at which thin structures are extracted. We indicate the known evaluation resolution after the method name. The results are:
> |Method|MonoSDF (Grid) -768|Ours -768|TUVR -1024|DebSDF (Grid) -1024|Ours -1024|DebSDF (Grid) -2048|Ours -2048|
> |:-:|:-:|:-:|:-:|:-:|:-:|:-:|:-:|
> |F-score↑|6.58|**9.35**|8.04|9.6|**14.46**|16.98|**28.96**|
>
> ND-SDF outperforms the baseline methods, especially at higher resolutions, indicating its superior ability to capture finer details and obtain more accurate reconstructions.
>
> The specific modifications of this part **in the revised version** are:
> - We update **Figure.4, Table.2, Table.3, Table.4**.
> - We update the corresponding experiment descriptions for these four indoor datasets in **Section.4.1**.
> - We add detailed comparison results on ScanNet++ and T&T in **Figure.15, Figure.16 and Figure.17**.
>
> We strongly suggest the reviewer to refer to the updated qualitative and quantitative comparison results.
>
> ## **About unbiasing methods**
>
> In Figure.3, we illustrate an actual scenario that the bias issue will affect the rendering weight. The main purpose of it is to stress the necessity of employing unbiasing transformation in SDF-based reconstructions. TUVR and DebSDF have presented detailed bias error curves in simpler scenarios, validating that the bias error follows the order: VolSDF ≥ Neus ≥ TUVR ≥ DebSDF. The key contribution of our method lies in identifying and addressing the unbiasing convergence issue, which is thoroughly analyzed in Appendix B.1, supported by both qualitative and quantitative ablation experiments.
>
> We further analyze the current bias issues in **Appendix C.4**; we strongly recommend that the reviewer refer to it for a more detailed discussion of these issues.
>
> ## **Advantages towards uncertainty based methods**
> H2OSDF uses a prior model to obtain uncertainty and then re-weights, suffering from limited generalization and robustness.
>
> In contrast to DebSDF's uncertainty field, we directly model the geometry-related SO(3) residuals to capture the deviation between prior geometry and scene geometry. Inherently, uncertainty is a kind of intensity field that is independent of geometry and measures the confidence of monocular cues. The variance term is introduced when using the log-likelihood loss, which is unrelated to geometry, may cause the model to focus on variance changes rather than the underlying data patterns.
>
> Our approach introduces a deflection term that is highly correlated with geometry, accurately reflecting deviations. DebSDF requires careful design of a variance threshold and tuning of numerous hyperparameters to prevent training instability, which may affect its generalization in complex scenes. We avoid this issue; the only aspect requiring special design in our method is the modulation function (Figure.8). Setting 𝑔(15°)=0.5 follows an intuitive design and yields promising results.
>
> ## **Mask comparisons and sampling strategy discussion**
> We visualize the masks obtained by various methods that have been mentioned in the review in Figure.12. And we provide a comprehensive analysis in Appendix B.3. They are now available **in the revised version**.
>
> ## **Trainig objective**
> For smooth regions, priors assist in rapid convergence during the initial warm-up phase. The designed $L^{ad}_{normal}$ loss provides a flexible capability, allowing the model to gradually converge in the right direction to fit the photometric consistency, even under supervision from incorrect priors. Since the gradients of modulation functions are detached, this indicates that the model autonomously learns the angular deviations of structural regions to converge correctly.

---

> ### Author Response · Authors · 2024-11-18
>
> ## **other questions**
> > what is the dataset conducted in the ablation study
>
> The ablation experiments in the main context are both conducted on the ScanNet++ dataset. Specifically, module ablation is performed on scene 036bce3393, and prior model ablation is carried out on scene 0e75f3c4d9 of ScanNet++.
>
> > The number of selected scenes in each dataset for the experiment.
>
> - For ScanNet, four scenes are selected, consistent with MonoSDF and DebSDF. We report it in Section 4.1 **ScanNet** paragraph.
> - For Replica, eight scenes are selected, consistent with MonoSDF and DebSDF.
> - For ScanNet++, six scenes are selected. The specific scene ids are summarized in Table.10
> - For T&T, we tested on the four indoor scenes of the advanced split, consistent with MonoSDF and DebSDF.
>
> > "GT depth" and "GT Normal"
>
> Thank you for carefully pointing out this problem! We have updated the framework figure (Figure.2) **in the revised version**.
>
> > MLP or Grid
>
> We have denoted the known scene representations of MonoSDF and DebSDF **in the revised version**.

---

> ### Comment · Reviewer_M33k · 2024-11-18
>
> The reviewer acknowledges the author's effort. All of my concerns have been addressed. The reviewer is glad to see the enhancement of this paper and decide to raise the rate from 5 to 6 and raise the confidence.
>
> Congratulations on the improvement.

---

> ### Author Response · Authors · 2024-11-18
>
> We are grateful to hear our feedback has addressed all of your concerns. It would be appreciated if you could recommend accepting our paper in the reviewers-pc discussion. We thank you again for your effort in reviewing our paper.
>
> Best regards,
>
> ND-SDF Authors

---

### Official Review · Reviewer_eHed · 2024-10-28

**Soundness:** 3
**Presentation:** 3
**Contribution:** 3
**Rating:** 8
**Confidence:** 5

**Summary:**

This paper proposes ND-SDF, which learns a normal deflection field to reduce the significant bias in monocular normal and depth priors. The normal deflection field can be rendered through volume rendering to further guide adaptive prior loss, ray sampling and unbiased rendering. The numerical and visualization results are compelling.

**Strengths:**

1. The inaccuracy and bias are widely existed in monocular depth and normal priors, therefore, the idea of exploring the uncertainty of monocular clues to improve the reconstruction performance is reasonable and valuable.
 2. The normal deflection field is effectively incorporated throughout the training process, enabling rendered deflected normal images to be supervised by estimated normals and adaptively applied in ray sampling, photometric loss, and unbiased rendering.
3. The numerical and visualization results are attractive and impressive, which highly demonstrate the effectiveness of the proposed method.
4. The ablation studies are thorough, clearly showcasing the effectiveness of each proposed module.

**Weaknesses:**

1. Lack of citation. A similar approach of learning rotated normals to mitigate the bias between normal priors and ground truth normals was first proposed in NC-SDF[1]. While I know that NC-SDF has not released the source code so it’s difficult to directly compare with it, an appropriate clarification is necessary to distinguish the proposed idea from NC-SDF.
2. Lack of quantitative comparisons with DebSDF. DebSDF also predicts the uncertainty of monocular priors and is able to reconstruct details such as chair legs. The numerical result of DebSDF is included in the table but visual comparison with DebSDF is absent, which is insufficient.

The methodology and experiments in this paper are solid. I like this paper, however, the two major concerns above prevent me from giving a higher score. I am willing to raise my score if the concerns are well addressed.

[1]. Chen Z, Wu X, Zhang Y. NC-SDF: Enhancing Indoor Scene Reconstruction Using Neural SDFs with View-Dependent Normal Compensation. Proceedings of the IEEE/CVF Conference on Computer Vision and Pattern Recognition. 2024: 5155-5165.

**Questions:**

See weakness.

---

> ### Author Response · Authors · 2024-11-18
>
> Thank you for the constructive comments and the valuable suggestions. Here, we address the concerns raised in the review.
>
> ## **Comparisons with newer methods**
> > Concerns about insufficient qualitative or quantitative comparison results with latest methods, such as TUVR, DebSDF.
>
> Additionally, we included DebSDF and TUVR for comparison. As TUVR is not open-sourced and was originally tested only on the DTU dataset, we incorporated TUVR's unbiasing function (Eq.15) into the official MonoSDF (Grid) framework to create the TUVR (MonoSDF-Grid) method, hereafter referred to simply as TUVR. The quantitative results are as follows:
>
> On Replica, we directly use the official evaluation results from DebSDF:
> |Method|TUVR|DebSDF (MLP)|Ours|
> |:-:|:-:|:-:|:-:|
> |F-score↑|85.81|88.36|**91.6**|
>
> On ScanNet++, our method also achieves the best results:
> |Method|TUVR|DebSDF (Grid)|Ours|
> |:-:|:-:|:-:|:-:|
> |F-score↑|53.7|69.6|**72.1**|
>
> For the T&T dataset, to our knowledge, DebSDF was evaluated at a 1024 resolution and MonoSDF at 768. To align with DebSDF, we increased the marching cubes resolution from 768 to 1024 and resubmitted it to the official T&T evaluation site. Additionally, we assessed comparative results at a higher resolution of 2048, at which more complex and thin structures are extracted. We indicate the known evaluation resolution after the method name. The results are:
> |Method|MonoSDF (Grid) -768|Ours -768|TUVR -1024|DebSDF (Grid) -1024|Ours -1024|DebSDF (Grid) -2048|Ours -2048|
> |:-:|:-:|:-:|:-:|:-:|:-:|:-:|:-:|
> |F-score↑|6.58|**9.35**|8.04|9.6|**14.46**|16.98|**28.96**|
>
> ND-SDF outperforms all baseline methods, especially at higher resolutions, indicating its superior ability to capture finer details and obtain more accurate reconstructions.
>
> The specific modifications of this part **in the revised version** are:
> - We update **Figure.4, Table.2, Table.3, Table.4**.
> - We update the corresponding experiment descriptions for these four indoor datasets in **Section.4.1**.
> - We add detailed comparison results on ScanNet++ and T&T in **Figure.15, Figure.16 and Figure.17**.
>
> We strongly suggest the reviewer to refer to the updated qualitative and quantitative comparison results.
>
> ## **Lack of citation**
>
> We clarify that our method is entirely independent of NC-SDF. Although we similarly model an SO(3) residual field to enhance surface reconstruction, our design diverges substantially. NC-SDF employs a two-stage approach, yet struggles with noise in the second stage’s explicit deflection. In contrast, our adaptive prior loss automatically assigns weights based on sample characteristics (Eq.12), removing the need for staged training. Additionally, NC-SDF requires extra boundary information for optimization, reducing robustness and adding hyperparameters. While NC-SDF was tested on datasets with simple layouts like ScanNet and ICL-NUIM, we evaluated our method on more complex indoor scenes, including ScanNet++ and T&T. Extensive experiments confirm our method’s superior generalization in challenging environments, an advantage not observed in NC-SDF.
>
> We updated this discussion **in Appendix C.4 in the revised version**.
>
>
> ## ***Optional: Confidence map comparisons**
> We visualize the masks obtained by various methods in Figure.12, including edge mask(EaNeuS), NCC mask(Equation 6 in NeuRIS), and uncertainty-aware mask(DebSDF). And we provide a comprehensive analysis in Appendix B.3. They are now available **in the revised version**.

---

> > ### Comment · Reviewer_eHed · 2024-11-18
> >
> > Thanks for the rebuttal. It addresses my major concerns. After reading the rebuttal and the discussions with other reviewers, I decide to raise my rating from 6 to 8.

---

> ### Author Response · Authors · 2024-11-18
>
> We are grateful to hear our feedback has addressed your major concerns. It would be appreciated if you could recommend accepting our paper in the reviewers-pc discussion. We thank you again for your effort in reviewing our paper.
>
> Best regards,
>
> ND-SDF Authors

---

### Official Review · Reviewer_KRgu · 2024-11-03

**Soundness:** 3
**Presentation:** 3
**Contribution:** 2
**Rating:** 8
**Confidence:** 4

**Summary:**

The paper proposed a method to improve prior normal-guided neural SDF indoor scene reconstruction. Instead of supervising the normal uniformly across the whole domain, the author proposes to use a network to learn the deflect normal and desired adaptive angle prior to loss according to the deviation of the deflect normal to the prior normal. The intuition is that large deviation happens in complex structure areas where the normal prior in general is bad quality.  The author also proposes to use the re-weighted loss on normal, depth, and color rendering.  The experiments show the proposed method performs well qualitatively and quantitatively.

**Strengths:**

1. The paper is well-written and easy to follow.
2. The normal deflection detection design is under reasonable assumptions and it offers a solution to deal with non-accurate normal prior.
3. The angle deviation loss design fits the situation.
4. The author provides complete experiments to show their method's performance. Including comparison with previous work quantitatively and qualitatively and ablation studies to show each module improves the results.

**Weaknesses:**

1. The bad quality normal prior problem seems only solved half. The author reduced the loss weight on the part that the normal prior is not accurate, but not the other half, what to do to improve the method itself performance in these areas.
2. Like the other prior-based methods, the quality of the prior matters. The author assumes smooth area the prior is relatively accurate.
3. Like the other SDF-based methods, the reconstruction seems to still struggle with the thin structures.
4. I noticed that in most cases the method outperforms previous methods, but sometimes it creates noise and artifacts.

**Questions:**

1. would it make sense to consider depth maps together to decide if the normal prior is accurate or not? For example, in most complex geometry areas such as thin structure areas, the depth is not continuous, and one may not be able to hope the normal is accurate there.
2. what is the reason for the artifacts (which I mentioned in weakness 4), is that because without supervision the model struggles to recover the details?
3. to deal with 2, maybe add some smooth loss / geometric preserving loss on these areas that the normal prior is categorized as bad?
4. I don't see a reason why this method only works indoor, has the author also tried it on outdoor datasets?

---

> ### Author Response · Authors · 2024-11-18
>
> Thank you for the thoughtful comments and constructive suggestions on our paper. Below, we address the concerns raised in the review.
>
> ## **Comparisons on DTU Dataset**
> ND-SDF's primary advantage is its capacity to handle scenes with complex layouts, encompassing both richly textured structures and large, smooth, low-textured areas. Generally, introducing blurred priors can reduce reconstruction accuracy, especially for textured objects or outdoor scenes. While our pipeline is not specifically tailored for single objects, we validated it on the DTU dataset without specifically adjusting parameters and reduced training steps from 128,000 to 50,000, yet still achieved promising results:
> |Method|Volsdf|Neus|MonoSDF (Grid)|Ours|
> |-|-|-|-|-|
> |CD↓|0.86|0.84|0.73|**0.66**|
>
> Notably, MonoSDF uses geometric priors only in the early training stages for quick initialization, avoiding potential accuracy loss from incorrect priors. In contrast, our approach consistently applies priors throughout the entire training process.
>
> ## **Noise and artifacts**
>
> There are indeed issues with noise and artifacts. From our opinion, the main cause of this is the poor quality of the ScanNet dataset and the use of hash grids for scene representation. The ScanNet dataset is captured by an iPad with an additional structure sensor. The images suffer from relatively low resolution, motion blur, and lighting variations. These poor-quality images led to significant perspective inconsistencies, which had a considerable impact on hashgrids, as they are locally structured for scene representation. Using an MLP would easily smooth out these issues, but in order to preserve as much detail as possible, we chose to use grids uniformly. Another cause are the non-uniform re-weight strategy (Section 3.4) guided by the deflection angles and the use of unbiasing method. But we believe their impact is minimal and controllable.
>
> To solve some reviews' concerns about insufficient qualitative and quantitative comparison results with newer methods. We add the latest state-of-the-art method DebSDF for comparison **in the revised version**. We add detailed comparison results on ScanNet++ and T&T in **Figure.15, Figure.16 and Figure.17**.
>
> The reviewer could refer to these new comparison results to check out the noise issue, by comparing our method with MonoSDF and DebSDF. If there are still concerns about significant artifacts, could the reviewer provide more detailed descriptions, such as which dataset, which figure, and the specific areas of concern?
>
> About the possible solutions of this problem, we have tried smooth term proposed in Permute-SDF and it has minimal improvement.
>
> ## **Other questions**
> > normal prior problem seems only solved half
>
> As stated in Section 3.3, we propose the $L_{normal}^{ad}$ (Eq.12) loss to adjust the utilization of various priors. We increase the weight ($g^d(\Delta\theta)$) of the deflected term (Eq.10) in structural regions to encourage the model to capture geometric deviations. In contrast, the weight ($g(\Delta\theta)$) of the naive normal prior term (Eq.5) decreases to prevent incorrect priors from interfering with fine-structure generation. In smooth regions, the deflected term's weight approaches zero, allowing the priors to fully guide the recovery of flat areas. Ideally, the adaptive loss ($L^{ad}_{normal}$) converges to zero. We consider this issue is well addressed. The reviewer can refer to Figure.8 for the $g^d,g$ curves.
> > An implausible assumption: smooth area the prior is relatively accurate.
>
> We believe this assumption is reasonably reliable. The monocular cues are generated by a pretrained neural network, which has a strong smooth inductive bias. Generally, smooth regions exhibit similar and simple feature patterns, enabling the network to more effectively capture and predict monocular cues.
> > Concerns about the poor ability of SDF methods to capture thin-structures
>
> We agree with this opinion since SDF methods inherently can not achieve very high-quality reconstructions, especially in large-scale indoor scenes with large-area smooth regions. But we still achieve remarkable reconstruction quality, balancing the smoothness and detail. The reviewer can refer to qualitative results in Appendix C for details (Figure 14-Figure 18).
>
> > Supervising using depth priors
>
> If the reviewer refers to designing an ad_depth structure similar to ad_normal, we believe this could introduce unnecessary noise into the normal deflection field learning. A more feasible approach might be to design a depth residual field. However, since depth priors lack a scale, the loss function may exhibit significant oscillations, leading to unstable convergence. This requires careful consideration. The ND-SDF learns the general difference between prior and scene geometries to distinguish structured regions, while simultaneously masking the depth priors. The effectiveness of the adaptive depth loss is validated in Appendix B.2.

---

> ### Comment · Reviewer_KRgu · 2024-11-18
> **Maintain the score**
>
> I thank the author for the detailed rebuttal. I do not have more questions. I'll maintain my score.

---

> ### Author Response · Authors · 2024-11-22
> **Additional Experiments and Insights Supporting Our Contributions**
>
> Thank you for your positive feedback. We are glad that our response has addressed the concerns raised in the review. However, we believe that certain points warrant further experimental validation for greater rigor.
>
> ## **Assumption: Priors of smooth regions are accurate**
> > Concerns about trivially assuming the high accuracy of monocular cues in simple flat regions.
>
> To deeply investigate this issue, we conducted experiments under the assumption that the priors for smooth regions are inaccurate. Specifically, we manually rotated all monocular normals in the smooth regions by 0°~60°. A SAM model was employed to detect the flat areas. The rotation was performed by rotating the normal priors by a specified degree (e.g., 60°) toward a pre-defined axis. We evaluated MonoSDF, DebSDF, and ND-SDF on the 0e75f3c4d scene from ScanNet++. The results are summarized as follows:
> |Methods|0°|5°|15°|30°|45°|60°| ***0°->60°***|
> |:-:|:-:|:-:|:-:|:-:|:-:|:-:|:-:|
> |Ours|0.750|0.735|0.678|0.654|0.709|0.655|**-0.095**|
> |MonoSDF|0.633|/|/|/|/|0.140|-0.493|
> |DebSDF|0.701|/|/|/|/|0.517|-0.184|
>
> As demonstrated above, decreases in F-score are observed across all methods. Specifically, MonoSDF produces an entirely abnormal result, as it rigidly relies on the biased priors. DebSDF fails to maintain robustness in this scenario. Although the uncertainty field in DebSDF can effectively mask irregular priors by learning uncertainty variances, it leads to inefficient utilization of the consistency information inherent in monocular cues within smooth regions. In contrast, our method explicitly models the actual bias amplitude, achieving notably robust performance with the smallest decrease in F-score, even when the normal priors are biased by as much as 60°. These findings further validate the generalization and robustness of our method in non-trivial assumptions of the monocular cues.
>
> We have included this study **in Appendix B.4 of the revised manuscript**, along with ***detailed visualization results***. The reviewer is encouraged to refer to this section for additional details.
>
> ## ***Optional: More comparison results**
> To solve some reviews' concerns about insufficient qualitative and quantitative comparison results with newer methods. We add the latest state-of-the-art method DebSDF for comparison on all these datasets in the **in the revised version**, including ScanNet, ScanNet++, Replica and T&T datasets .
>
> The specific modifications of this part are:
> - We update **Figure.4, Table.2, Table.3, Table.4**.
> - We update the corresponding experiment descriptions for these four indoor datasets in **Section.4.1**.
> - We add detailed comparison results on ScanNet++ and T&T in **Figure.16, Figure.17 and Figure.18**.
>
> We demonstrate the superiority of our method to capture accurate detailed structures, even in large-scale or complex indoor scenes. The updated comparisons prove the better generalization abilities of our method than DebSDF.
>
> ## ***Optional: Confidence map comparisons**
> We visualize the masks that highlight structural regions obtained by various methods in Figure.12, including edge mask(EaNeuS), NCC mask(Equation 6 in NeuRIS), and uncertainty-aware mask(DebSDF). And we provide a comprehensive analysis in Appendix B.3. They are now available **in the revised version**.
>
> ## **End**
> We hope our reply could provide the reviewer with deeper insights into our approach. As the discussion phase is nearing its end, we would be grateful to hear your feedback and wondered if you might still have any concerns we could address. It would be appreciated if you could raise your score on our paper. We thank you again for your effort in reviewing our paper.
>
> Best regards,
>
> ND-SDF Authors

---

> > ### Comment · Reviewer_KRgu · 2024-11-26
> > **Thank you for your clarification**
> >
> > Thank you for your detailed reply. I believe the paper's contribution is good and I certainly vote to accept the paper.
> >
> > I mention that the normal problem is half-solved in that the paper leverages the good normal prior part and a lower weight for the non-smooth normal part (which is reasonable). It would be another large improvement to have some other prior that could also be used or another method to estimate good-quality normal in these regions. And hopefully, solve the thin structure problem.
> > It is not a criticism of the paper since I understand it is out of the scope of this paper.
> >
> > I thank the authors for the hard work again and I'll raise my score to 8.

---

> ### Author Response · Authors · 2024-11-26
> **Response to reviewer KRgu**
>
> ## ***Optional: Additional details about the normal problem**
> We truly thank the reviewer for clarifying the normal problem. Actually, this problem can be related to our training objective:
> > **Training objective of ND-SDF**: For smooth regions, priors assist in rapid convergence during the initial warm-up phase (Appendix A.2). The designed $L^{ad}_{normal}$ loss (Eq.12) provides a flexible capability, allowing the model to gradually converge in the right direction to fit the photometric consistency and generate accurate geometry, even when supervised with incorrect priors in structural regions. Since the gradients of modulation functions are detached, this suggests that the model autonomously learns the angular deviations of structural regions to converge correctly.
>
> The above indicates that the photometric loss, rather than the suppressed normal loss, is mainly utilized to remedy the structural regions. The specially designed re-weight, deviations-driven sampling, and partial unbiasing strategies are for further boosting the recovery of these regions. Indeed, we designed exactly these strategies to address the half-normal problem. Maybe other high-quality priors can be directly used to guide the optimization of thin structures as the reviewer mentioned.
>
> ## **End**
> Thank you for your positive comments on our work. We truly appreciate your acknowledgment of our contributions and the raised score. Once again, we thank you for your time and effort in reviewing our paper.
>
> Best regards,
>
> ND-SDF Authors

---

### Official Review · Reviewer_EWRb · 2024-11-04

**Soundness:** 3
**Presentation:** 3
**Contribution:** 3
**Rating:** 8
**Confidence:** 3

**Summary:**

This paper presents a pipeline for multi-view 3D reconstruction of indoor scenes. The pipeline is based on recently popular differentiable volume rendering methods using Signed Distance Functions (SDF), such as VolSDF and NeuS. To enhance reconstruction quality in indoor scenes, particularly in textureless regions, the paper introduces monocular priors for additional supervision, inspired by recent works like MonoSDF and NeuRIS.

A core contribution of this paper is the introduction of a normal deflection field, which helps the model identify and correct inaccuracies in reference normal images estimated from monocular models, such as the Omnidata model. Experimental results show that this deflection field aids in reconstructing fine geometric details, such as thin structures.

Additionally, the authors introduce techniques to further improve geometry reconstruction quality, including (1) an adaptive normal loss weight and (2) unbiased volume rendering on thin objects.

The experimental results are promising and include comprehensive ablation studies. Overall, the paper is well-argued and substantiated.

**Strengths:**

- The proposed normal deflection fields, which correct inaccurately estimated normals from normal maps, seem reasonable to me.
- Table 6 and Figure 5 effectively showcase the contribution of the method's design through comprehensive ablation studies.
- The experimental results demonstrate that this proposed pipeline achieves state-of-the-art reconstruction quality in indoor scenarios.

**Weaknesses:**

- Using unbiased volume rendering for thin structures seems reasonable. However, why not apply this unbiased volume rendering weight uniformly across all regions? The authors mention potential convergence issues with this approach, but it would be helpful if they provided a more in-depth analysis explaining the nature of these convergence issues and offered insights into why they occur.

**Questions:**

- In Line 280, is this a typo on $g(\theta)$? Looks like this is a missing $\Delta$.

---

> ### Author Response · Authors · 2024-11-18
> **Kind Reminder to Review Our Rebuttal**
>
> Thank you for your thoughtful feedback. Below, we address the concerns raised in the review.
>
> ## **Convergence issue**
> > We found that uniformly applying the unbiasing transformation (Eq. 15) can cause convergence issues. To address this, we propose a partial unbiasing strategy, or deflection angle-guided unbiased rendering (Eq. 27).
>
> This issue has been analyzed in **Appendix B.1** in the original paper. And we present both qualitative and quantitative ablation experiments to validate it. We believe the reviewer may have missed the relevant information.
>
> Besides, we additionally visualize the inverse variance (1/β) of the SDF-density conversion function (Eq.3) during training and find that, without partial unbiasing, 1/β fails to converge to a relatively high value, indicating inaccurate and blurry surfaces. The variance curves and comprehensive analysis are presented **in the revised PDF version**; reviewers can refer to  Appendix B.1 for details.
>
> We add a very detailed discussion **in Appendix C.4** about the unbiasing method and the motivation of the partial unbiasing strategy. If the reviewers are interested, refer to this section for details.
>
> ## ***Optional: More comparison results**
> To solve some reviews' concerns about insufficient qualitative and quantitative comparison results with newer methods. We add the latest state-of-the-art method DebSDF for comparison on all these datasets in the **in the revised version**, including ScanNet, ScanNet++, Replica and T&T datasets .
>
> The specific modifications of this part are:
> - We update **Figure.4, Table.2, Table.3, Table.4**.
> - We update the corresponding experiment descriptions for these four indoor datasets in **Section.4.1**.
> - We add detailed comparison results on ScanNet++ and T&T in **Figure.16, Figure.17 and Figure.18**.
>
> We demonstrate the superiority of our method to capture accurate detailed structures, even in large-scale or complex indoor scenes. The updated comparisons prove the better generalization abilities of our method than DebSDF.
>
> ## ***Optional: Confidence map comparisons**
> We visualize the masks obtained by various methods in Figure.12, including edge mask(EaNeuS), NCC mask(Equation 6 in NeuRIS), and uncertainty-aware mask(DebSDF). And we provide a comprehensive analysis in Appendix B.3. They are now available **in the revised version**.
>
> ## **Answer to questions**
> In Line 280, the $\theta$ is correctly changed to $\Delta\theta$. Thank you for carefully pointing out this problem!

---

> ### Author Response · Authors · 2024-11-22
>
> We hope our reply could address your questions. As the discussion phase is nearing its end, we would be grateful to hear your feedback and wondered if you might still have any concerns we could address. It would be appreciated if you could raise your score on our paper. We thank you again for your effort in reviewing our paper.
>
> Best regards,
>
> ND-SDF Authors

---

> > ### Comment · Reviewer_EWRb · 2024-11-27
> > **Thank you**
> >
> > Thank you authors for the detailed responses. This addressed my concerns. I'll raise my score.

---

> > > ### Author Response · Authors · 2024-11-27
> > > **Response to reviewer EWRb**
> > >
> > > We are grateful to hear our feedback has addressed your concerns. We sincerely appreciate your recognition of our contributions and the increased score. Thank you once again for your time and effort in reviewing our paper.
> > >
> > > Best regards,
> > >
> > > ND-SDF Authors

---

### Author Response · Authors · 2024-11-18

We sincerely appreciate the reviewers' time and effort in reviewing our work. Thanks for raising these valuable questions and comments, as they have motivated us to further validate our findings. We will gladly incorporate all the feedback in the revised version. Here we address some common concerns:
## **Comparisons with newer methods**
> Concerns about insufficient qualitative or quantitative comparison results with latest methods, such as TUVR, DebSDF.

Additionally, we included DebSDF and TUVR for comparison. As TUVR is not open-sourced and was originally tested only on the DTU dataset, we incorporated TUVR's unbiasing function (Eq.15) into the official MonoSDF (Grid) framework to create the TUVR (MonoSDF-Grid) method, hereafter referred to simply as TUVR. The quantitative results are as follows:

On Replica, we directly use the official evaluation results from DebSDF:
|Method|TUVR|DebSDF (MLP)|Ours|
|:-:|:-:|:-:|:-:|
|F-score↑|85.81|88.36|**91.6**|

On ScanNet++, our method also achieves the best results:
|Method|TUVR|DebSDF (Grid)|Ours|
|:-:|:-:|:-:|:-:|
|F-score↑|53.7|69.6|**72.1**|

For the T&T dataset, to our knowledge, DebSDF was evaluated at a 1024 resolution and MonoSDF at 768. To align with DebSDF, we increased the marching cubes resolution from 768 to 1024 and resubmitted it to the official T&T evaluation site. Additionally, we assessed comparative results at a higher resolution of 2048, at which more complex and thin structures are extracted. We indicate the known evaluation resolution after the method name. The results are:
|Method|MonoSDF (Grid) -768|Ours -768|TUVR -1024|DebSDF (Grid) -1024|Ours -1024|DebSDF (Grid) -2048|Ours -2048|
|:-:|:-:|:-:|:-:|:-:|:-:|:-:|:-:|
|F-score↑|6.58|**9.35**|8.04|9.6|**14.46**|16.98|**28.96**|

ND-SDF outperforms all baseline methods, especially at higher resolutions, indicating its superior ability to capture finer details and obtain more accurate reconstructions.

The above experiments also shows that simply using the unbiased TUVR method does not achieve satisfactory reconstruction results. We further visualize the detailed comparison results with DebSDF on ScanNet++ and T&T datasets **in the revised version**; reviewers can refer to Appendix.C for more information.
## **Comparisons on DTU Dataset**
ND-SDF's primary advantage is its capacity to handle scenes with complex layouts, encompassing both richly textured structures and large, smooth, low-textured areas. Generally, introducing blurred priors can reduce reconstruction accuracy, especially for textured objects or outdoor scenes. While our pipeline is not specifically tailored for single objects, we validated it on the DTU dataset without specifically adjusting parameters and reduced training steps from 128,000 to 50,000, yet still achieved promising results:
|Method| Volsdf  |  Neus  | MonoSDF (Grid)  |  Ours |
|:-:|:-:|:-:|:-:|:-:|
|CD↓|0.86|0.84|0.73|**0.66**|

Notably, MonoSDF uses geometric priors only in the early training stages for quick initialization, avoiding potential accuracy loss from incorrect priors. In contrast, our approach consistently applies priors throughout the entire training process.

## **Convergence issue**
> We found that uniformly applying the unbiasing transformation (Eq. 15) can cause convergence issues. To address this, we propose a partial unbiasing strategy, or deflection angle-guided unbiased rendering (Eq. 27).

This issue has been analyzed in Appendix B.1, where we present both qualitative and quantitative ablation experiments to validate it. Reviewers who may have missed this can refer to the section for details.

We further visualize the inverse variance (1/β) of the SDF-density conversion function (Eq.3) during training and find that, without partial unbiasing, 1/β fails to converge to a relatively high value, indicating inaccurate and blurry surfaces. The variance curves and a thorough analysis are presented **in the revised version**; reviewers can refer to  Appendix B.1 for details.

## **Additional discussions**
> Lack of citation: the difference between ND-SDF and NC-SDF

We clarify that our method is entirely independent of NC-SDF. Although we similarly model an SO(3) residual field to enhance surface reconstruction, our design diverges substantially. NC-SDF employs a two-stage approach, yet struggles with noise in the second stage’s explicit deflection. In contrast, our adaptive prior loss automatically assigns weights based on sample characteristics (Eq.12), removing the need for staged training. Additionally, NC-SDF requires extra boundary information for optimization, reducing robustness and adding hyperparameters. While NC-SDF was tested on datasets with simple layouts like ScanNet and ICL-NUIM, we evaluated our method on more complex indoor scenes, including ScanNet++ and T&T. Extensive experiments confirm our method’s superior generalization in challenging environments, an advantage not observed in NC-SDF.

---

### Author Response · Authors · 2024-11-27
***Optional for all reviewers: Update on Supplementary Materials**

## **Supplementary Material update**
We have updated the anonymous homepage of the supplementary materials by replacing all baseline methods for slider interaction visualizations with DebSDF. Additionally, we have included visualizations of textured meshes generated by our method on ScanNet++. Reviewers can download the **updated supplementary materials** for further details.

## **End**
We want to express our sincere gratitude to all the reviewers for their thoughtful and constructive feedback. Your insightful suggestions and detailed comments have significantly contributed to the improvement of our work. We truly appreciate your acknowledgment of our contributions. Thank you again for your time and effort in reviewing our manuscript.

Best regards,

Authors

---

### Meta-Review · Area_Chair_ysLq · 2024-12-14

**Metareview:**

The paper presents an approach to multi-view 3D reconstruction of indoor scenes via neural implicit functions and priors from monocular depth and surface estimation. To adaptively balance the influence of priors for reconstructing finer details, the paper proposes to learn a normal deflection field, which models the difference between normals from the prior and those from a pure image-based reconstruction. To make use of the learned field during optimization the paper further proposes losses and a sampling strategy. The efficacy of the proposed techniques is validated in an ablation study and comparisons against the state of the art. The reviewers unanimously recommended to accept the paper and the AC agrees with this assessment.

**Additional Comments On Reviewer Discussion:**

The discussion resolved all open questions.

---

### Decision · Program_Chairs · 2025-01-22

Accept (Spotlight)